# Train Once, Get a Family: State-Adaptive Balances for Offline-to-Online Reinforcement Learning

**Shenzhi Wang**[1*], **Qisen Yang**[1*], **Jiawei Gao**[1], **Matthieu Lin**[2], **Hao Chen**[4], **Liwei Wu**[1]
**Ning Jia**[3], **Shiji Song**[1], **Gao Huang**[1†]
[1] Department of Automation, BNRist, Tsinghua University
[2] Department of Computer Science, BNRist, Tsinghua University
[3] Beijing Academy of Artificial Intelligence (BAAI)    [4] Independent Researcher
Project Page: `https://shenzhi-wang.github.io/NIPS_FamO2O`

## Abstract

Offline-to-online reinforcement learning (RL) is a training paradigm that combines pre-training on a pre-collected dataset with fine-tuning in an online environment. However, the incorporation of online fine-tuning can intensify the well-known distributional shift problem. Existing solutions tackle this problem by imposing a *policy constraint* on the *policy improvement* objective in both offline and online learning. They typically advocate a single balance between policy improvement and constraints across diverse data collections. This one-size-fits-all manner may not optimally leverage each collected sample due to the significant variation in data quality across different states. To this end, we introduce Family Offline-to-Online RL (FamO2O), a simple yet effective framework that empowers existing algorithms to determine state-adaptive improvement-constraint balances. FamO2O utilizes a *universal model* to train a family of policies with different improvement/constraint intensities, and a *balance model* to select a suitable policy for each state. The-oretically, we prove that state-adaptive balances are necessary for achieving a higher policy performance upper bound. Empirically, extensive experiments show that FamO2O offers a statistically significant improvement over various existing methods, achieving state-of-the-art performance on the D4RL benchmark. Codes are available at `https://github.com/LeapLabTHU/FamO2O`.

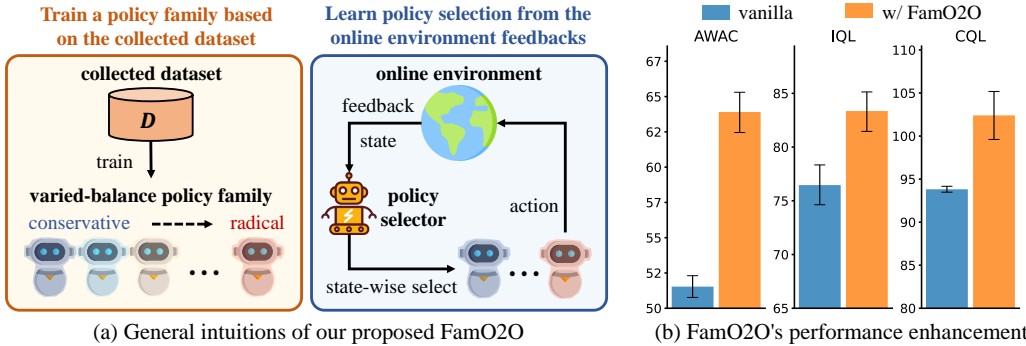

(a) General intuitions of our proposed FamO2O          (b) FamO2O's performance enhancement

Figure 1: FamO2O trains a policy family from datasets and selects policies state-adaptively using online feedback. Easily integrated, FamO2O statistically enhances existing algorithms' performance.

---

[*] Equal contribution. Email: {`wsz21`, `yangqs19`}`@mails.tsinghua.edu.cn`.
[†] Corresponding author. Email: `gaohuang@tsinghua.edu.cn`.

37th Conference on Neural Information Processing Systems (NeurIPS 2023).

# 1 Introduction

Offline reinforcement learning (RL) provides a pragmatic methodology for acquiring policies utilizing pre-existing datasets, circumventing the need for direct environment interaction [29]. Nonetheless, the attainable policy performance in offline RL is frequently constrained by the quality of the dataset [66]. The offline-to-online RL paradigm addresses this limitation by refining the offline RL policy through fine-tuning in an online setting [35].

While online fine-tuning can indeed elevate policy performance, it also potentially exacerbates the issue of distributional shift [29], where policy behavior diverges from the dataset distribution. Such shifts typically ensue from drastic policy improvements and are further amplified by state distribution changes when transitioning from offline learning to online fine-tuning [11, 25, 9, 26]. Prior works have attempted to counter this by imposing policy constraints on the policy improvement objective to deter excessive exploration of uncharted policy space [25, 54, 35]. However, this conservative approach can inadvertently stifle policy improvement [35]. In essence, offline-to-online RL necessitates an effective balance between *policy improvement* and *policy constraint* during policy optimization.

Regrettably, prior offline-to-online RL algorithms tend to adopt a monolithic approach towards this improvement-constraint trade-off, indiscriminately applying it to all data in a mini-batch [10, 66, 53] or the entire dataset [35, 24, 27, 30]. Given the inherent data quality variation across states (see Figure 2), we argue that this one-size-fits-all manner may fail to optimally exploit each sample. In fact, data yielding high trajectory returns should encourage more "conservative" policies, while data leading to poor returns should incite more "radical" policy improvement.

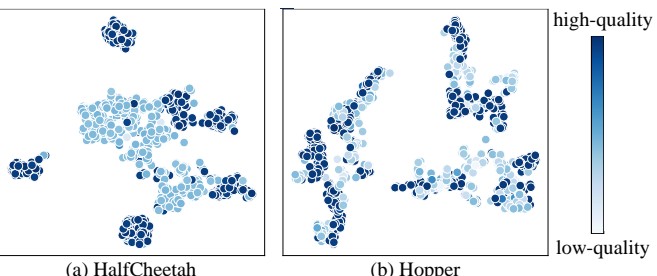

(a) HalfCheetah    (b) Hopper

Figure 2: A t-SNE [46] **visualization of randomly selected states** from (a) HalfCheetah and (b) Hopper medium-expert datasets in D4RL [8]. The color coding represents the return of the trajectory associated with each state. This visualization underscores **the significant variation in data quality across different states**.

In this paper, we introduce a novel framework, Family Offline-to-Online RL (FamO2O), which can discern a state-adaptive improvement-constraint balance for each state. FamO2O's design is founded on two key insights delineated in Figure 1(a): (i) The collected dataset, abundant in environmental information, could facilitate the training of a diverse policy family ranging from conservative to radical, and (ii) feedback from online learning might assist in selecting an appropriate policy from this family for each state. As depicted in Figure 3, FamO2O incorporates a *universal model* and a *balance model*. The universal model, conditioned on a *balance coefficient*, determines the degree of policy conservatism or radicalism, while the balance model learns a mapping from states to balance coefficients, aiding the universal model in tailoring its behavior to each specific state.

FamO2O represents, to the best of our knowledge, the first offline-to-online RL approach harnessing the power of state-adaptive improvement-constraint balances. Theoretically, we establish that in policy optimization, *point-wise KL constraints* afford a superior performance upper bound compared to the *distributional KL constraints* adopted in prior works [37, 35, 24]. Importantly, these state-adaptive balances become indispensable when addressing point-wise KL constraints, thereby underlining the necessity of incorporating such balances in offline-to-online RL. Experimental results, as summarized in Figure 1(b), reveal that FamO2O is a simple yet effective framework that statistically significantly improves various offline-to-online RL algorithms and achieves state-of-the-art performance.

# 2 Preliminaries

We introduce essential RL and offline-to-online RL concepts and notations here. For descriptive convenience and theoretical analysis, we use the advantaged-weight regression (AWR) algorithm framework [37, 35, 24], but **FamO2O isn't limited to the AWR framework**. We later demonstrate its integration with non-AWR algorithms in Section 5.3 and Appendix D.

## 2.1 Reinforcement learning formulation

RL is typically expressed as a Markov decision process (MDP) [43], denoted as $(\mathcal{S}, \mathcal{A}, P, d_0, R, \gamma)$. Here, $\mathcal{S}$ and $\mathcal{A}$ are the state[1] and action space; $P(\mathbf{s}_{t+1}|\mathbf{s}_t, \mathbf{a})$ is the environmental state transition probability; $d_0(\mathbf{s}_0)$ represents initial state distribution; $R(\mathbf{s}_t, \mathbf{a}_t, \mathbf{s}_{t+1})$ is the reward function; and $\gamma \in (0, 1]$ is the discount factor.

## 2.2 Offline-to-online reinforcement learning

Offline-to-online RL is a training paradigm including two phases: (i) *offline pre-training*: pre-training a policy based on an offline dataset; (ii) *online fine-tuning*: fine-tuning the pre-trained policy by interacting with the environment. Note that in the offline pre-training phase, the policy cannot interact with the environment, but in the online fine-tuning phase, the policy has access to the offline dataset.

Similar to offline RL algorithms, offline-to-online RL algorithms' training objectives usually consist of two terms, either explicitly [35, 24] or implicitly [27, 28]: (i) *policy improvement*, which aims to optimize the policy according to current value functions; (ii) *policy constraint*, which keeps the policy around the distribution of the offline dataset or current replay buffer. Using the AWR algorithm framework [37, 35, 24], we demonstrate our method (also applicable to non-AWR algorithms) and define notations in Equation (1) for later use.

$$L_\pi = \mathbb{E}_{(\mathbf{s},\mathbf{a})\sim\mathcal{D}} \left[ \overbrace{\exp\Big( \underbrace{\beta}_{\text{balance coefficient}} \big( \underbrace{Q(\mathbf{s},\mathbf{a}) - V(\mathbf{s})}_{\text{policy improvement}} \big) \Big)}^{\text{imitation weight}} \cdot \underbrace{\log \pi(\mathbf{a}|\mathbf{s})}_{\text{policy constraint}} \right]. \tag{1}$$

$L_\pi$ is a maximization objective, while $\mathcal{D}$ represents the collected dataset. Initially, during offline pre-training, $\mathcal{D}$ starts with the pre-collected offline dataset $\mathcal{D}_{\text{offline}} = \{(\mathbf{s}_k, \mathbf{a}_k, \mathbf{s}'_k, r_k) \mid k = 1, 2, \cdots, N\}$. As we move to the online fine-tuning phase, online interaction samples are continuously incorporated into $\mathcal{D}$ [35, 24]. The balance coefficient $\beta$ is a predefined hyperparameter moderating between the policy improvement and policy constraint terms, while the imitation weight sets the imitation intensity for the state-action pair $(s, a)$.

## 3 State-Adaptive Balance Coefficients

Our design of state-adaptive improvement-constraint balances is motivated by the observations that (i) the quality of the dataset's behavior, *i.e.*, the trajectory returns, fluctuates greatly with different states, as shown in Figure 2; (ii) state-dependent balances are conducive to a higher performance upper bound. In this section, we will theoretically validate the latter point.

We first present the policy optimization problem with point-wise KL constraints in Definition 3.1, which is the focus of FamO2O:

**Definition 3.1** (Point-wise KL Constrained Optimization Problem). We consider a policy optimization problem defined as follows:

$$\max_\pi \ \mathbb{E}_{\mathbf{s}\sim d\pi_\beta(\cdot), \mathbf{a}\sim\pi(\cdot|\mathbf{s})} \left[ Q^{\pi^k}(\mathbf{s},\mathbf{a}) - V^{\pi^k}(\mathbf{s}) \right] \tag{2}$$

$$\text{s.t. } D_{\text{KL}}(\pi(\cdot|\mathbf{s})|\pi_\beta(\cdot|\mathbf{s})) \leq \epsilon_\mathbf{s}, \quad \forall \mathbf{s} \in \mathcal{S} \tag{3}$$

$$\int_{\mathbf{a}\in\mathcal{A}} \pi(\mathbf{a}|\mathbf{s})\, \mathrm{d}\mathbf{a} = 1, \quad \forall \mathbf{s} \in \mathcal{S}. \tag{4}$$

Here, $\pi^k(k \in \mathbb{N})$ denotes the policy at iteration $k$, $\pi_\beta$ signifies a behavior policy representing the action selection way in the collected dataset $\mathcal{D}$, $d_{\pi_\beta}(\mathbf{s})$ refers to the state distribution of $\pi_\beta$, and $\epsilon_\mathbf{s}$ is a state-related constant. The optimal policy derived from Equations (2) to (4) is designated as $\pi^{k+1}$.

The optimization problem common in previous work [37, 35, 24] is shown in Definition 3.2:

**Definition 3.2** (Optimization problem with distributional KL constraints). The definition of the policy optimization problem with distributional KL constraints is the same as Definition 3.1, except that

---

[1]For simplicity, we use "state" and "observation" interchangeably in fully or partially observed environments.

Equation (3) in Definition 3.1 is substituted by Equation (5), where $\epsilon$ is a constant:

$$\int_{\mathbf{s}\in\mathcal{S}} d_{\pi_\beta}(\mathbf{s}) D_{\mathrm{KL}}(\pi(\cdot|\mathbf{s})\|\pi_\beta(\cdot|\mathbf{s}))\, \mathrm{d}\mathbf{s} \leq \epsilon. \tag{5}$$

*Remark* 3.3. The update rule in Equation (1) is based on the optimization problem in Definition 3.2.

The point-wise constraints' superiority over distributional constraints is shown in Proposition 3.4:

**Proposition 3.4** (Advantage of point-wise KL constraints). *Denote the optimal value in Definition 3.1 as $J_*^k[\{\epsilon_\mathbf{s}, \mathbf{s}\in\mathcal{S}\}]$, the optimal value in Definition 3.2 as $J_*^k[\epsilon]$. These optimal values satisfy:*

$$\forall \epsilon \geq 0, \quad \exists\{\epsilon_\mathbf{s}, \mathbf{s}\in\mathcal{S}\}, \quad J_*^k[\{\epsilon_\mathbf{s}, \mathbf{s}\in\mathcal{S}\}] \geq J_*^k[\epsilon]. \tag{6}$$

*Proof.* Please refer to Appendix C.1. $\qquad\qquad\qquad\qquad\qquad\qquad\qquad\qquad\qquad\square$

Proposition 3.4 indicates that the optimal value under the point-wise KL constraints, given suitable point-wise constraints, is *no less than* that under distributional KL constraints. This finding justifies our approach under point-wise constraints.

Proposition 3.5 shows the necessity of state-dependent balance coefficient design in solving the point-wise KL constraint optimization problem:

**Proposition 3.5** (State-dependent balance coefficient). *Consider the optimization problem in Definition 3.1. Assume that the state space $\mathcal{S} = [s_{\min}, s_{\max}]^l$ ($l$ is the state dimension), and the feasible space constrained by Equations (3) to (4) is not empty for every $\mathbf{s}\in\mathcal{S}$. Then the optimal solution of $\pi^{k+1}$, denoted as $\pi_*^{k+1}$, satisfies that $\forall\mathbf{s}\in\mathcal{S}, \mathbf{a}\in\mathcal{A}$,*

$$\pi_*^{k+1}(\mathbf{a}|\mathbf{s}) = \frac{\pi_\beta(\mathbf{a}|\mathbf{s})}{Z_\mathbf{s}} \exp\left(\beta_\mathbf{s}(Q^{\pi^k}(\mathbf{s},\mathbf{a}) - V^{\pi^k}(\mathbf{s}))\right), \tag{7}$$

*where $\beta_\mathbf{s}$ is a state-dependent balance coefficient, and $Z_\mathbf{s}$ is a normalization term. When utilizing a parameterized policy $\pi_\phi$ to approximate the optimal policy $\pi_*^{k+1}$, the training objective can be formulated as:*

$$\phi = \arg\max_\phi \mathbb{E}_{(\mathbf{s},\mathbf{a})\sim\mathcal{D}}\left[\exp(\beta_\mathbf{s}(Q^{\pi^k}(\mathbf{s},\mathbf{a}) - V^{\pi^k}(\mathbf{s})))\log\pi_\phi(\mathbf{a}|\mathbf{s})\right]. \tag{8}$$

*Proof.* The proof is deferred to Appendix C.2. $\qquad\qquad\qquad\qquad\qquad\qquad\qquad\qquad\square$

In contrast to AWR [37] and AWAC [35], Proposition 3.5 highlights state-dependent (marked in blue) balance coefficients in Equations (7) to (8), as opposed to a pre-defined hyperparameter in Equation (1). This state-adaptiveness is due to Proposition 3.5 considering the finer-grained constraints in Definition 3.1. Together, Proposition 3.4 and Proposition 3.5 indicate state-adaptive balance coefficients contribute to a higher performance upper bound.

## 4 Family Offline-to-Online RL

Section 3 theoretically proves that finer-grained policy constraints enhance performance upper bounds, necessitating state-adaptive balance coefficients. Accordingly, we introduce FamO2O, a framework adaptively assigning a balance coefficient to each state, easily implemented over various offline-to-online algorithms like [35, 24, 27], hereafter called the "base algorithm".

Essentially, FamO2O trains a policy family with varying balance coefficients during offline pre-training. During online fine-tuning, FamO2O

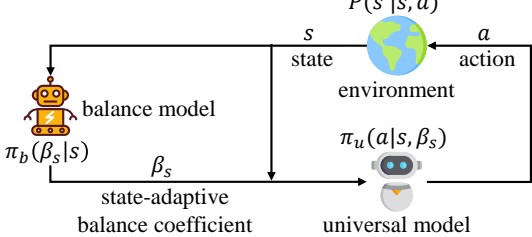

Figure 3: FamO2O's **inference** process. For each state $\mathbf{s}$, the balance model $\pi_b$ computes a state-adaptive balance coefficient $\beta_\mathbf{s}$. Based on $\mathbf{s}$ and $\beta_\mathbf{s}$, the universal model $\pi_u$ outputs an action $\mathbf{a}$.

identifies the appropriate policy, corresponding to the suitable balance coefficient, for each state from this policy family. In this section, we present FamO2O using the AWR algorithm framework (Equation (1)). FamO2O's compatibility with non-AWR algorithms is discussed in Appendix D.

As shown in Figure 3, FamO2O's policy consists of two components: a *universal model* $\pi_u$ and a *balance model* $\pi_b$. Denote the space of balance coefficients as $\mathcal{B}$. For every state $\mathbf{s}$, the balance model[2] $\pi_b : \mathcal{S} \mapsto \mathcal{B}$ figures out a suitable balance coefficient $\beta_\mathbf{s}$; based on the state $\mathbf{s}$ and state-related balance coefficient $\beta_\mathbf{s}$, the universal model $\pi_u : \mathcal{S} \times \mathcal{B} \mapsto \mathcal{A}$ outputs an action. The balance coefficient $\beta_\mathbf{s}$ is to control the conservative/radical degree of the universal model $\pi_u$ in dealing with the state $\mathbf{s}$.

### 4.1 Learning universal model

We initially address training the universal model $\pi_u$, aimed at learning a policy family with varying balances between policy improvement and constraint. The formal optimization target of $\pi_u$ is:

$$\pi_u^{k+1} = \arg\max_{\pi_u} \mathbb{E}_{(\mathbf{s},\mathbf{a})\sim\mathcal{D}} \Big[ \exp(\beta_\mathbf{s}(Q^k(\mathbf{s},\mathbf{a}) - V^k(\mathbf{s}))) \log \pi_u(\mathbf{a}|\mathbf{s}, \beta_\mathbf{s}) \Big]. \tag{9}$$

$Q^k$ and $V^k$, detailed in Section 4.3, represent $Q$ and $V$ functions at iteration $k$. Equation (9) echoes Equation (8), but the policy also takes a balance coefficient $\beta_\mathbf{s}$ as input (highlighted in blue). In the offline pre-training phase, $\beta_\mathbf{s}$ is randomly sampled from balance coefficient space $\mathcal{B}$. This encourages $\pi_u$ to learn varied strategies. During online fine-tuning, $\beta_\mathbf{s}$ is set by balance model $\pi_b$ before input to universal model $\pi_u$, which prompts cooperation between $\pi_u$ and $\pi_b$.

### 4.2 Learning balance model

Next, we outline how the balance model $\pi_b$ chooses an appropriate policy for each state from the policy family trained by the universal model $\pi_u$. As indicated in Section 4.2, every $\beta_\mathbf{s} \in \mathcal{B}$ corresponds to a unique policy. Consequently, to select the optimal policy, $\pi_b$ needs to determine the appropriate balance coefficient $\beta_\mathbf{s}$ for each state $\mathbf{s}$. Given this rationale, the update rule for $\pi_b$ is:

$$\pi_b^{k+1} = \arg\max_{\pi_b} \mathbb{E}_{(\mathbf{s},\mathbf{a})\sim\mathcal{D}} \Big[ Q^k(\mathbf{s}, \underbrace{\pi_u^{k+1}(\mathbf{s}, \overbrace{\pi_b(\mathbf{s})}^{\text{balance coefficient } \beta_\mathbf{s}})}_{\text{action}}) \Big]. \tag{10}$$

Here, $\pi_u^{k+1}$ represents the updated universal model in Equation (9). Intuitively, Equation (10) aims to find a $\pi_b$ that maximizes $Q^k$ value by translating balance coefficients into actions with $\pi_u^{k+1}$. This design is grounded in the understanding that the $Q$ value serves as an estimate of future return, which is our ultimate goal of striking a balance between policy improvement and constraint. Concerns may arise about $Q^k$'s extrapolation error in Equation (10) potentially misguiding $\pi_b$'s update. Empirical evidence suggests this is less of an issue if we avoid extremely radical values in the balance coefficient space $\mathcal{B}$. Following the update rule in Equation (10), $\pi_b$ effectively assigns balance coefficients to states, demonstrated in Section 6.1.

### 4.3 Learning value functions

Furthermore, we explain the value functions update. As per Equations (9) to (10), a single set of $Q$ and $V$ functions evaluate both $\pi_u$ and $\pi_b$. This is due to $\pi_b : \mathcal{S} \mapsto \mathcal{B}$ and $\pi_u : \mathcal{S} \times \mathcal{B} \mapsto \mathcal{A}$ collectively forming a standard RL policy $\pi_u(\cdot, \pi_b(\cdot)) : \mathcal{S} \mapsto \mathcal{A}$. Hence, the value functions update mirrors that in the base algorithm, simply replacing the original policy with $\pi_u(\cdot, \pi_b(\cdot))$.

Finally, we offer a pseudo-code of FamO2O's training process in Appendix B.

## 5 Experimental Evaluation

In this section, we substantiate the efficacy of FamO2O through empirical validation. We commence by showcasing its state-of-the-art performance on the D4RL benchmark [8] with IQL [24] in Section 5.1. We then evidence its performance improvement's generalizability and statistical significance in Section 5.2. Moreover, Section 5.3 reveals FamO2O's compatibility with non-AWR-style algorithms like CQL [27], yielding significant performance enhancement. Lastly, we reserve detailed FamO2O analyses for Section 6 and Appendix F. For more information on implementation details, please refer to Appendix E.2.

---

[2]Despite $\pi_u$ and $\pi_b$ being stochastic models, we notate them as functions using "$\mapsto$" hereafter for brevity.

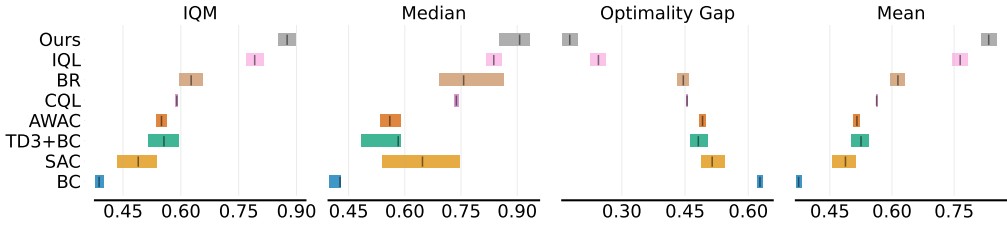

Figure 5: **Comparisons between our FamO2O against various competitors** on D4RL normalized scores [8]. All methods are tested on D4RL Locomotion and AntMaze for 6 random seeds. FamO2O achieves state-of-the-art performance by a statistically significant margin among all the competitors in offline-to-online RL (*i.e.* IQL [24], Balaned Replay (BR) [28], CQL [27], AWAC [35], and TD3+BC [10]), online RL (*i.e.* SAC [13]), and behavior cloning (BC).

Table 1: **Enhanced performance achieved by FamO2O after online fine-tuning.** We evaluate the D4RL normalized score [8] of standard base algorithms (including AWAC [35] and IQL [24], denoted as "Base") in comparison to the base algorithms augmented with FamO2O (referred to as "Ours"). All results are assessed across 6 random seeds. The superior offline-to-online scores are highlighted in blue. **FamO2O consistently delivers statistically significant performance enhancements across different algorithms and task sets.**

| Dataset[1] | AWAC [35] | | IQL [24] | | Avg. | |
|---|---|---|---|---|---|---|
| | Base | Ours | Base | Ours | Base | Ours |
| hopper-mr-v2 | 56.0 | 86.8 | 91.0 | 97.6 | 73.5 | 92.2 |
| hopper-m-v2 | 54.1 | 75.0 | 65.4 | 90.7 | 59.7 | 82.8 |
| hopper-me-v2 | 97.7 | 92.9 | 76.5 | 87.3 | 87.1 | 90.1 |
| halfcheetah-mr-v2 | 43.9 | 49.0 | 53.7 | 53.1 | 48.8 | 51.0 |
| halfcheetah-m-v2 | 44.8 | 47.6 | 52.5 | 59.2 | 48.7 | 53.4 |
| halfcheetah-me-v2 | 91.0 | 90.6 | 92.8 | 93.1 | 91.9 | 91.8 |
| walker2d-mr-v2 | 72.8 | 84.4 | 90.1 | 92.9 | 81.5 | 88.6 |
| walker2d-m-v2 | 79.0 | 80.0 | 83.8 | 85.5 | 81.4 | 82.8 |
| walker2d-me-v2 | 109.3 | 108.5 | 112.6 | 112.7 | 110.9 | 110.6 |
| **locomotion total** | 648.4 | 714.9 | 718.3 | 772.0 | 683.4 | 743.4 |
| **95% CIs** | 640.5~656.8 | 667.3~761.4 | 702.5~733.5 | 753.5~788.5 | 674.6~692.0 | 732.1~754.2 |
| umaze-v0 | 64.0 | 96.9 | 96.5 | 96.7 | 80.4 | 96.8 |
| umaze-diverse-v0 | 60.4 | 90.5 | 37.8 | 70.8 | 66.2 | 80.6 |
| medium-diverse-v0 | 0.2 | 22.2 | 92.8 | 93.0 | 45.2 | 57.6 |
| medium-play-v0 | 0.0 | 34.2 | 91.5 | 93.0 | 45.2 | 63.6 |
| large-diverse-v0 | 0.0 | 0.0 | 57.5 | 64.2 | 24.7 | 32.1 |
| large-play-v0 | 0.0 | 0.0 | 52.5 | 60.7 | 21.4 | 30.3 |
| **antmaze total** | 124.7 | 243.7 | 428.7 | 478.3 | 283.1 | 361.1 |
| **95% CIs** | 116.5~132.6 | 226.2~259.9 | 406.7~452.7 | 456.7~498.7 | 274.1~291.1 | 347.2~374.3 |
| **total** | 773.0 | 958.6 | 1146.9 | 1250.3 | 960.0 | 1104.5 |
| **95% CIs** | 761.5~784.6 | 936.8~979.6 | 1119.5~1175.1 | 1221.9~1277.0 | 911.3~1008.8 | 1063.5~1145.4 |

[1] mr: medium-replay, m: medium, me: medium-expert.

**Datasets** Our method is validated on two D4RL [8] benchmarks: Locomotion and AntMaze. Locomotion includes diverse environment datasets collected by varying quality policies. We utilize IQL [24] settings, assessing algorithms on hopper, halfcheetah, and walker2d environment datasets, each with three quality levels. AntMaze tasks involve guiding an ant-like robot in mazes of three sizes (umaze, medium, large), each with two different goal location datasets. The evaluation environments are listed in Table 1's first column.

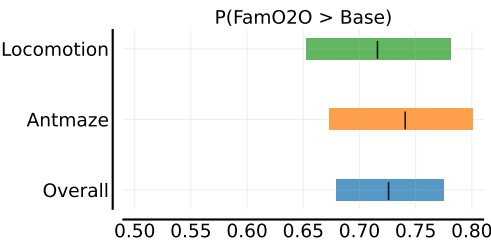

Figure 4: **FamO2O's improvement over base algorithms [35, 24].** For D4RL Locomotion, AntMaze [8], and overall, FamO2O shows significant and meaningful performance gains, meeting Neyman-Pearson criteria [3].

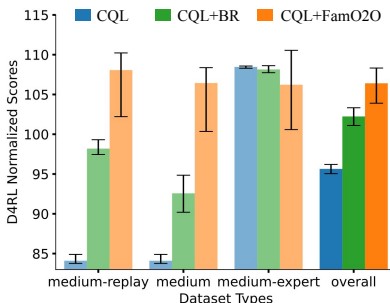
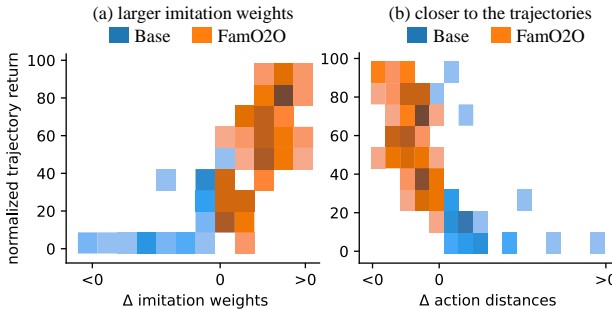

Figure 6: **IQM scores of FamO2O's implementation on top of CQL [27].** Demonstrating a statistically significant superiority over vanilla CQL [27] and CQL+BR [28], FamO2O affirms its **adaptability to non-AWR algorithms**.

Figure 7: Comparing FamO2O with the base algorithm [24] on **(a) imitation weights** and **(b) action distances**. Δ indicates the difference between FamO2O's metrics and the base's. Generally, FamO2O **emphasizes high-quality data imitation** and **aligns more closely with high-quality trajectories** compared to the base algorithm.

**Metrics**    Considering RL's inherent variability, we adopt robust evaluation methods per rliable [3]. Besides conventional Medium and Mean, we integrate IQM and Optimality Gap metrics for broader assessment. We also employ rliable's probability of improvement metric for gauging the likelihood of our method outperforming others. We confirm our performance enhancement's statistical significance using 95% Confidence Intervals (CIs).

## 5.1    Benchmark Comparison

FamO2O's state-of-the-art performance is demonstrated by implementing it over IQL [24] and comparing with baseline methods.

**Baselines**    We benchmark FamO2O against: (i) **offline-to-online RL**, including IQL [24], Balanced Replay (BR) [28], CQL [27], AWAC [35], and TD3+BC [10]. For IQL, BR, and AWAC, official implementations are used. In the case of CQL and TD3+BC, we implement online fine-tuning based on the author-provided offline pre-training codes, following the procedures in IQL and AWAC; (ii) **online RL method**, SAC [13], to highlight offline pre-training's efficacy; (iii) **behavior cloning (BC)**, which is implemented by maximizing the log-likelihood of the samples in the offline dataset. For SAC and BC, we utilize the implementations of CQL. Further details are in Appendix E.1.

**Comparison**    As shown in Figure 5, FamO2O outperforms competitors across all metrics (IQM, Medium, Optimality Gap, and Mean). Specifically, for IQM, Optimality Gap, and Mean, FamO2O's 95% CIs don't overlap with the competitors'. Even for Medium, all baseline expectations fall below the lower limit of FamO2O's 95% CIs. The results underscore the significant edge of our state-adaptive policy constraint mechanism over competing methods.

## 5.2    Analyzing FamO2O's Performance Enhancement

Though FamO2O demonstrated superior performance on the D4RL benchmark in Section 5.1, it's vital to discern the actual contributions of FamO2O from its base policy, IQL. Therefore, we address two key questions: (i) Does FamO2O consistently enhance other offline-to-online RL algorithms? (ii) Is the performance boost by FamO2O statistically significant given RL's inherent variability?

**Setup**    We apply FamO2O to AWAC and IQL. AWAC [35] is one of the most famous offline-to-online algorithms, and IQL [24] is a recently proposed method that achieves great performance on D4RL [8]. We use the authors' codes, maintaining the same hyperparameters for a fair comparison. Further details are in Appendix E.2.

**Comparison**    Table 1 shows AWAC's and IQL's performances w/ and w/o FamO2O. FamO2O generally enhances performance by a statistically significant margin across most datasets, regardless of the base algorithm, highlighting its versatility. Even on complex datasets where AWAC barely succeeds, *e.g.*, AntMaze medium-diverse and medium-play, FamO2O still achieves commendable performance.

Pursuing rliable's recommendation [3], we evaluated FamO2O's statistical significance by calculating average probabilities of improvement against base policies (Figure 4). In all three cases (Locomotion,

AntMaze, and Overall), the lower CI bounds exceed 50%, denoting the statistical significance of FamO2O's improvement. Specifically, the upper CI on Locomotion surpasses 75%, demonstrating statistical meaning as per the Neyman-Pearson criterion.

### 5.3 Versatility of FamO2O with Non-AWR Algorithms

To demonstrate FamO2O's versatility beyond AWR-based algorithms, we extended it to CQL [27] in addition to AWAC [35] and IQL [24]. The implementation specifics are in Appendix D. As Figure 6 reveals, FamO2O significantly outperforms CQL. Even when compared to Balance Replay (BR) [28], an offline-to-online method designed specifically for CQL, FamO2O still shows statistically significant superior performance. These results highlight FamO2O's adaptability to non-AWR algorithms.

## 6 Discussion

In this section, we further provide some in-depth studies on FamO2O, including visualization (Section 6.1) and quantitative analyses (Sections 6.2 to 6.6). More analyses are deferred to Appendix F.

### 6.1 Does FamO2O really have state-wise adaptivity?

Here, we design a simple maze environment to visualize the state-wise adaptivity of FamO2O. As shown in Figure 8(a), the agent starts at a random cell in the top row and is encouraged to reach the goal at the bottom right corner through two crossing points. During data collection, guidance is provided when the agent passes through the lower crossing point, but no guidance for the upper crossing point. To elaborate, the guidance refers to compelling the agent to adhere to the route and direction that yields the shortest path to the goal. Without it, the agent moves randomly. It can be observed in Figure 8(b) that the agent generally outputs lower balance coefficients for the states with high-quality samples (*i.e.*, those derived from the agent's movement with guidance) while generating higher

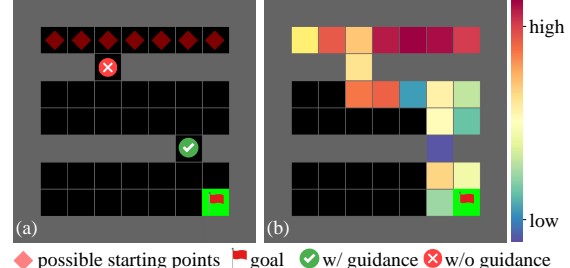

◆ possible starting points ⚑ goal ✔ w/ guidance ✖ w/o guidance

Figure 8: **State-wise adaptivity visualization** in a simple maze environment. (a) Higher data quality at the crossing point in the 5th row compared to the 2nd row. (b) Colors denote different balance coefficient values at traversed cells during inference. FamO2O typically displays conservative (or radical) behavior at cells with high-quality (or low-quality) data.

balance coefficients for the states with low-quality data (*i.e.*, data gathered when the agent moves without guidance). This result shows FamO2O's state-wise adaptivity in choosing proper balance coefficients according to the data quality related to the current state.

### 6.2 What is the effect of state-adaptive balances?

In this section, we explore the impact of state-adaptive improvement-constraint balance. In order to encompass data of varying quality levels, we assess FamO2O using the medium-replay datasets of D4RL Locomotion [8]. Our analysis focuses on two metrics: *imitation weights* and *action distances*.

*Imitation weights* are defined in Equation (1), with larger (or smaller) values prompting the agent to align more closely (or less) with the replay buffer $\mathcal{D}$'s behavior. *Action distance*, delineated in Equation (11), quantifies the discrepancy in action selection between a policy $\pi$ and a trajectory $\tau$:

$$d_{\text{action}}^{\pi,\tau} = \mathbb{E}_{(\mathbf{s},\mathbf{a})\sim\tau}\left[\|\arg\max_{\mathbf{a}'}\pi(\mathbf{a}'|\mathbf{s}) - \mathbf{a}\|_2^2\right]. \tag{11}$$

Here, a lower (or higher) action distance $d_{\text{action}}^{\pi,\tau}$ signifies a greater (or lesser) behavioral similarity between the policy $\pi$ and the trajectory $\tau$.

We evaluate IQL with and without FamO2O ('Base' and 'FamO2O') regarding imitation weights and action distances, depicted in Figure 7. Figure 7(a) computes the average imitation weight difference (AIWD) per trajectory in the offline dataset. AIWD indicates the mean imitation weight difference between FamO2O and the base algorithm for each $(s, a)$ pair within a trajectory. Figure 7(b) likewise determines an average action distance difference per offline dataset trajectory.

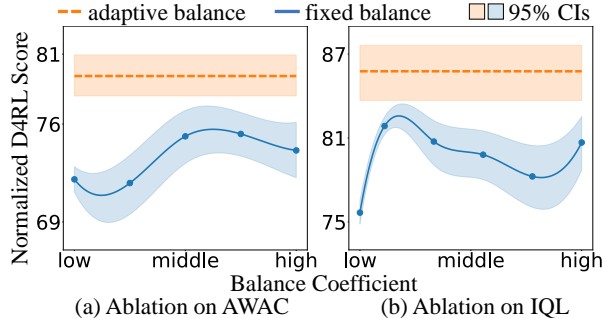
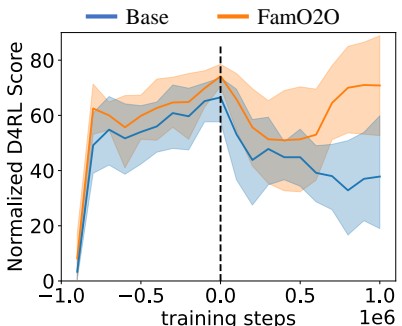

(a) Ablation on AWAC     (b) Ablation on IQL

Figure 9: **Comparing adaptive and fixed balance coefficients** on (a) AWAC [35] and (b) IQL [24] using D4RL Locomotion datasets [8]. Adaptive coefficients consistently outperform fixed ones with distinct 95% CIs.

Figure 10: FamO2O **alleviates performance drop** due to **distributional shift** during the shift from offline pre-training to online fine-tuning.

Figure 7(a) reveals that FamO2O typically shows higher imitation weights than the base algorithm for high-return trajectories. Figure 7(b) indicates that FamO2O aligns more with high-quality trajectories and less with low-quality ones than the base algorithm. These results highlight the state-adaptive balance's role in promoting emulation of high-quality behavior and avoidance of low-quality behavior.

### 6.3 State-adaptive balances *vs.* fixed balances?

To prove that our adaptive balance coefficients are better than traditional fixed balance coefficients, we compare FamO2O against the base algorithms with different balance coefficients as hyperparameters. As shown in Figure 9, on both AWAC [35] and IQL [24], our adaptive balance coefficients outperform all the fixed balance coefficients. Significantly, the 95% CIs of adaptive and fixed balance coefficients have no overlap. These comparison results indicate that our adaptive balance coefficient approach surpasses the fixed balance coefficient method by a statistically significant margin.

### 6.4 Does FamO2O's efficacy rely on varied data qualities?

It's worth noting that our method's efficacy doesn't rely on varied data qualities. Table 1 clearly demonstrates that FamO2O surpasses the base algorithms in performance across all the medium datasets, namely hopper-medium-v2, halfcheetah-medium-v2, and walker2d-medium-v2, which all maintain a relatively consistent data quality. We claim that FamO2O can determine the suitable conservative/radical balance for each state in online scenarios based on the data quality in the collected dataset. If the dataset is diverse in quality, the balances will be diverse; if the quality is consistent, the balances will be correspondingly consistent. The above claim is supported by Figure 11, which indi-

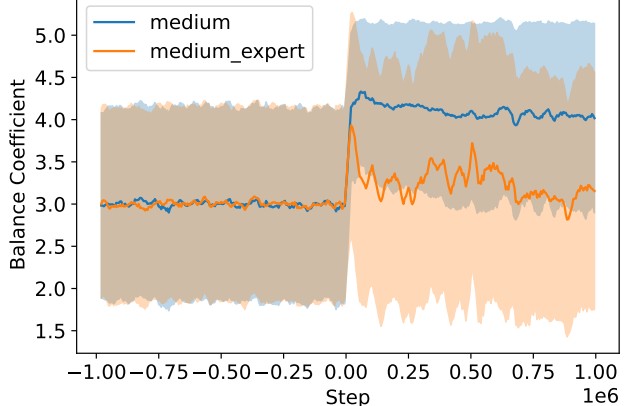

Figure 11: Balance coefficients' mean and std for IQL w/ FamO2O on D4RL HalfCheetah.

cates that (i) in datasets with more (or less) diverse data qualities, i.e., medium-expert (or medium), the balance coefficients are more (or less) diverse, with corresponding larger (or smaller) standard deviations; (ii) with higher (or lower) quality datasets, the balance coefficients are averagely lower (or higher), leading to a more conservative (or radical) policy.

### 6.5 Does FamO2O mitigate the performance drop stemming from the distributional shift?

Despite FamO2O's primary objective not being direct distributional shift handling, its state-adaptive improvement-constraint balances prove beneficial in mitigating performance degradation during the

offline pre-training to online fine-tuning transition, attributed to the distributional shift. Existing offline-to-online RL algorithms [24, 35] already incorporate mechanisms to counter distributional shift, hence significant performance drops are infrequent. Figure 10 illustrates the training curves of IQL and IQL+FamO2O across both offline pre-training (negative steps) and online fine-tuning (positive steps) on antmaze-umaze-diverse, on which IQL exhibits the most significant decline in performance when transitioning from offline pre-training to online fine-tuning. As evidenced, while IQL+FamO2O initially experiences a performance drop akin to IQL during online fine-tuning, it recovers rapidly and attains high performance, in stark contrast to IQL's sustained performance decline throughout the fine-tuning stage.

### 6.6 Balance model *vs.* random selector?

To validate the effect of the balance model $\pi_b$ in choosing balance coefficients, we present a FamO2O variant, denoted as *random-FamO2O*, with the balance model replaced with a random balance coefficient selector. Other training settings keep the same for FamO2O and random-FamO2O. Table 2 shows the improvement percentages and probability of improvement of FamO2O against random-FamO2O. As we can observe, FamO2O outperforms random-FamO2O on almost all the datasets of D4RL Locomotion [8]. Furthermore, the lower CI of the probability of improvement is much higher than 50%, and the upper CI exceeds 75%. This indicates that the effect of the balance model is not only statistically significant but also statistically meaningful as per the Neyman-Person statistical testing criterion.

Table 2: **Ablation study on the balance model.** The absence of the balance model results in decreased performance of FamO2O on nearly all D4RL Locomotion datasets [8]. The model's impact is statistically significant and meaningful as per the Neyman-Person criterion.

| Dataset | FamO2O | random-FamO2O |
|---|---|---|
| hopper-mr-v2 | 97.64 | 80.87 |
| hopper-m-v2 | 90.65 | 86.37 |
| hopper-me-v2 | 87.28 | 77.08 |
| halfcheetah-mr-v2 | 53.07 | 53.75 |
| halfcheetah-m-v2 | 59.15 | 53.15 |
| halfcheetah-me-v2 | 93.10 | 92.72 |
| walker2d-mr-v2 | 92.85 | 91.38 |
| walker2d-m-v2 | 85.50 | 84.84 |
| walker2d-me-v2 | 112.72 | 110.54 |
| **total** | **771.96** | **730.70** |
| **prob. of improvement** | **0.70** (95% CIs: **0.59~0.79**) | |

## 7 Conclusion

This work underscores the significance of state-adaptive improvement-constraint balances in offline-to-online RL. We establish a theoretical framework demonstrating the advantages of these state-adaptive balance coefficients for enhancing policy performance. Leveraging this analysis, we put forth Family Offline-to-Online RL (FamO2O), a versatile framework that equips existing offline-to-online RL algorithms with the ability to discern appropriate balance coefficients for each state.

Our experimental results, garnered from a variety of offline-to-online RL algorithms, offer substantial evidence of FamO2O's ability to significantly improve performance, attaining leading scores on the D4RL benchmark. In addition, we shed light on FamO2O's adaptive computation of state-adaptive improvement-constraint balances and their consequential effects through comprehensive analyses. Ablation studies on the adaptive balances and balance model further corroborate FamO2O's efficacy.

The limitation of our work is that FamO2O has been evaluated on just a handful of representative offline-to-online RL algorithms, leaving a vast array of such algorithms unexamined. Additionally, FamO2O's utility is somewhat limited, as it is applicable exclusively to offline-to-online algorithms. In future work, we aim to expand the applicability of FamO2O by integrating it with a broader spectrum of offline-to-online algorithms. Additionally, we intend to investigate the feasibility of incorporating state-adaptive improvement-constraint balances in offline RL settings, where online feedback is intrinsically absent, or even applying our adaptive design to the LLM-as-agent domain [49, 50].

## Acknowledgement

This work was supported in part by the Key-Area Research and Development Program of Guangdong Province under Grant 2020B1111500002, the National Natural Science Foundation of China under Grants 62022048 and 62276150, and the Guoqiang Institute of Tsinghua University.

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

# Appendices

## Contents

## A Related Work

**Offline RL** Reinforcement learning (RL) usually improves a policy through continuous online interactions with the environment [43, 32, 62]. To reduce the huge demand for online interactions, especially when they are costly or risky, researchers proposed offline RL that utilizes pre-collected datasets to improve the policy without interacting with the environment [29]. Directly applying off-policy algorithms in offline RL usually leads to poor performance. This phenomenon is due to *distributional shift*, where the agent may learn inaccurate value estimates of out-of-distribution (OOD) state-action pairs [29, 61, 60]. Existing algorithms address this issue via policy constraints [11, 25, 54, 23, 37, 35], importance sampling [19, 14, 65], regularization [33, 27], uncertainty estimation [2], and imitation learning [4, 42, 52].

**Offline-to-online RL** Offline-to-online RL adds online fine-tuning to offline RL to enhance policy performance. Similar to the distributional shift problem in offline RL, offline-to-online (O2O) RL also suffers from off-policy bootstrapping error accumulation caused by OOD data, which causes a large "dip" in the initial performance of online fine-tuning [35]. Moreover, O2O RL algorithms tend to be excessively conservative and result in plateauing performance improvement [35]. This emphasizes the importance of finding a balance between being optimistic about improving the policy during the online phase and still being constrained to the conservative offline policy. However, as previously discussed, existing O2O RL algorithms generally adopt a "one-size-fits-all" trade-off, either in the process of Q-learning [27, 30, 24, 31], along with the policy improvement [35], via a revised replay buffer [28], or through some alignment approaches [53, 59]. It's worth noting that, [66] also proposes an adaptive weight in online fine-tuning. However, the adaptive weight is a single parameter for all the samples. In addition, its update requires an extra human-defined target score, which is absent in the setting considered by our competitors and us. Furthermore, [63] presents a policy expansion strategy that adaptively selects a policy from a set, which includes both offline pre-trained policies and online learned policies. Nevertheless, this policy set comprises only two policies, in contrast to the infinite number of policies contained in FamO2O's policy family.

**Conditioned RL policy** Algorithms in this scheme can be divided into two stages: (i) a set of policies is trained; (ii) the policy with the highest performance is selected. Generally, these methods learn a family of policies that exhibit different behaviors and search for the best one that successfully executes the desired task [36, 57]. A similar method to our FamO2O is UP-OSI [58], where a Universal Policy (UP) within a parameterized space of dynamic models and an Online System Identification (OSI) function that predicts dynamic model parameters based on recent state and action history are proposed. Unlike these prior methods utilizing models to identify the environmental parameters, the balance model in our method aims to determine a proper balance between policy improvement and constraints for each state. It's worth noting that [18, 45, 12] also exhibit conditioned designs. However, [18] focuses on conditioned value functions and pure offline RL; [45] relies on human interactions; [12] concentrates on adjusting offline RL policies according to the estimated probabilities of various MDPs in uncovered areas, but uncoverage might not be a primary concern in offline-to-online RL due to the accessibility to the online environment. Moreover, [56, 55] both exhibit a comparable understanding of how to adjust policy improvement and policy constraint across samples of varying quality. Nevertheless, the focus of [56, 55] is on offline RL. Specifically, [56] requires a small number of expert demonstrations, while [55] necessitates annotated preferences to appropriately calibrate the balance for each individual sample. Additionally, in areas such as computer vision, there are existing works that condition their training and/or inference on various input samples [15, 16, 40, 38, 39, 51]. These studies align with the adaptive approach in our work.

**Hierarchical RL** Hierarchical Reinforcement Learning (HRL) decomposes complex tasks into manageable subtasks, enhancing sample efficiency, interpretability, and generalization in RL. HRL approaches bifurcate into predefined and learned options (or skills). Predefined options, guided by domain knowledge or human demonstrations, shape high-level policy or constrain action space, exemplified by MAXQ [6], options framework [44], skill chaining [22], and feudal RL [5]. Learned options, optimizing objectives like entropy or reward, are seen in option-critic, skill-discovering [22], feudal networks [48], HEXQ [17], DIAYN [7], HIRO [34], and HIDIO [64]. While FamO2O shares some design elements with HRL, such as hierarchical policy decision-making, the objectives diverge: FamO2O focuses on balancing policy improvement and constraints per state, while HRL seeks to simplify complex tasks via goals or options.

## B   Training Process of FamO2O

Here we provide the pseudo-codes of FamO2O (see Algorithm 1) to demonstrate its training process.

---

**Algorithm 1** FamO2O Algorithm

---

**Require:** replay buffer $\mathcal{D}$, offline dataset $\mathcal{D}_{\text{offline}}$, offline and online training steps $N_{\text{off}}, N_{\text{on}}$
1: Initialize $\pi_b^0, \pi_u^0, Q^0, V^0$
2: Initialize $\mathcal{D}$ with $\mathcal{D}_{\text{offline}}$
3: **for** $k = 0 \rightarrow N_{\text{off}} - 1$ **do**                          ▷ offline pre-training phase
4:      Sample a mini-batch $M_k$ from $\mathcal{D}$
5:      Assign a random balance coefficient for each sample in $M_k$, denoting the balance coefficient set as $B_{M_k}$
6:      Update $\pi_u^k$ to $\pi_u^{k+1}$ with $M_k, B_{M_k}$ by Equation (9)
7:      Update $\pi_b^k$ to $\pi_b^{k+1}$ with $M_k$ by Equation (10)
8:      Update $Q^k, V^k$ to $Q^{k+1}, V^{k+1}$ respectively with $M_k$ according to the base algorithm
9: **end for**
10: **for** $k = N_{\text{off}} \rightarrow N_{\text{on}} - 1$ **do**                          ▷ online fine-tuning phase
11:      Collect samples with $\pi_u^k, \pi_b^k$ and add samples to $\mathcal{D}$
12:      Sample a mini-batch $M_k$ from $\mathcal{D}$
13:      $\pi_b^k$ computes a balance coefficient for each sample in $M_k$. Denote the balance coefficient set as $B_{M_k}$
14:      Update $\pi_u^k$ to $\pi_u^{k+1}$ with $M_k, B_{M_k}$ by Equation (9)
15:      Update $\pi_b^k$ to $\pi_b^{k+1}$ with $M_k$ by Equation (10)
16:      Update $Q^k, V^k$ to $Q^{k+1}, V^{k+1}$ respectively with $M_k$ according to the base algorithm
17: **end for**

---

# C  Theoretical Proofs

## C.1  Proof of Proposition 3.4

**Lemma C.1.** *Equation* (5) *is equivalent to:*

$$\exists\{\epsilon_{\mathbf{s}}, \mathbf{s} \in \mathcal{S}\} \in C, \quad \forall \mathbf{s} \in \mathcal{S}, \quad D_{\mathrm{KL}}(\pi(\cdot|\mathbf{s})\|\pi_\beta(\cdot|\mathbf{s})) \le \epsilon_{\mathbf{s}}, \tag{12}$$

$$where\ C = \left\{ \{\epsilon_{\mathbf{s}}', \mathbf{s} \in \mathcal{S}\} \,|\, \int_{\mathbf{s} \in \mathcal{S}} d_{\pi_\beta}(\mathbf{s})\epsilon_{\mathbf{s}}' \,\mathrm{d}\mathbf{s} = \epsilon, \epsilon_{\mathbf{s}}' \ge 0 \right\}. \tag{13}$$

*Proof.* We aim to prove Equation (5) $\Rightarrow$ Equations (12) to (13) in our manuscript through a contradiction approach. Let's assume that Equation (5) $\not\Rightarrow$ Equations (12) to (13), signifying the following:

1. **Condition 1**  According to Equation (5), $\int_{\mathbf{s} \in \mathcal{S}} d_{\pi_\beta(\mathbf{s})} D_{\mathrm{KL}}(\pi(\cdot|\mathbf{s})\|\pi_\beta(\cdot|\mathbf{s}))\mathrm{d}\mathbf{s} \le \epsilon$;

2. **Condition 2**  The converse proposition of Equations (12) to (13) is: $\forall \epsilon_{\mathbf{s}} \ge 0$ ($\mathbf{s} \in \mathcal{S}$) which satisfy $\int_{\mathbf{s} \in \mathcal{S}} d_{\pi_\beta}(\mathbf{s})\epsilon_{\mathbf{s}}\mathrm{d}\mathbf{s} = \epsilon$, there exists $\mathbf{s} \in \mathcal{S}$, $D_{\mathrm{KL}}(\pi(\cdot|\mathbf{s})\|\pi_\beta(\cdot|\mathbf{s})) > \epsilon_{\mathbf{s}}$.

Next, we form a special set $\{\epsilon_{\mathbf{s}}, \mathbf{s} \in \mathcal{S}\}$. Selecting an arbitrary $\mathbf{s}_0 \in \mathcal{S}$, the set fulfills:

1. For all $\mathbf{s} \ne \mathbf{s}_0$, $\epsilon_{\mathbf{s}} = D_{\mathrm{KL}}(\pi(\cdot|\mathbf{s})\|\pi_\beta(\cdot|\mathbf{s}))$,

2. $\epsilon_{\mathbf{s}_0} = \epsilon - \int_{\mathbf{s} \in \mathcal{S}, \mathbf{s} \ne \mathbf{s}_0} d_{\pi_\beta}(\mathbf{s}) D_{\mathrm{KL}}(\pi(\cdot|\mathbf{s}_0)\|\pi_\beta(\cdot|\mathbf{s}_0))\mathrm{d}\mathbf{s}$.

It can be clearly seen that the set adheres to $\int_{\mathbf{s} \in \mathcal{S}} d_{\pi_\beta}(\mathbf{s})\epsilon_{\mathbf{s}}\mathrm{d}\mathbf{s} = \epsilon$. Based on Condition 2, we can deduce that $D_{\mathrm{KL}}(\pi(\cdot|\mathbf{s}_0)\|\pi_\beta(\cdot|\mathbf{s}_0)) > \epsilon_{\mathbf{s}_0}$. Hence,

$$
\begin{aligned}
\epsilon =& \int_{\mathbf{s} \in \mathcal{S}} d_{\pi_\beta}(\mathbf{s})\epsilon_{\mathbf{s}}\mathrm{d}\mathbf{s} \\
=& \ d_{\pi_\beta}(\mathbf{s}_0)\epsilon_{\mathbf{s}_0} + \int_{\mathbf{s} \in \mathcal{S}, \mathbf{s} \ne \mathbf{s}_0} d_{\pi_\beta}(\mathbf{s})\epsilon_{\mathbf{s}}\mathrm{d}\mathbf{s} \\
<& \ d_{\pi_\beta}(\mathbf{s}_0)D_{\mathrm{KL}}(\pi(\cdot|\mathbf{s}_0)\|\pi_\beta(\cdot|\mathbf{s}_0)) + \int_{\mathbf{s} \in \mathcal{S}, \mathbf{s} \ne \mathbf{s}_0} d_{\pi_\beta}(\mathbf{s})D_{\mathrm{KL}}(\pi(\cdot|\mathbf{s})\|\pi_\beta(\cdot|\mathbf{s}))\mathrm{d}\mathbf{s} \\
=& \int_{\mathbf{s} \in \mathcal{S}} d_{\pi_\beta}(\mathbf{s})D_{\mathrm{KL}}(\pi(\cdot|\mathbf{s})\|\pi_\beta(\cdot|\mathbf{s}))\mathrm{d}\mathbf{s}.
\end{aligned}
$$

This stands in contradiction to condition 1, where $\int_{\mathbf{s} \in \mathcal{S}} d_{\pi_\beta(\mathbf{s})} D_{\mathrm{KL}}(\pi(\cdot|\mathbf{s})\|\pi_\beta(\cdot|\mathbf{s}))\mathrm{d}\mathbf{s} \le \epsilon$. Thus, the assumption that Equation (5) $\not\Rightarrow$ Equations (12) to (13) is proven false, which confirms that Equation (5) $\Rightarrow$ Equations (12) to (13). $\qquad\square$

Denote the feasible region of the problem in Definition 3.2 as $\mathcal{F}^k(\epsilon)$, and the feasible region of the problem in Definition 3.1 as $\mathcal{F}^k(\{\epsilon_{\mathbf{s}}, \mathbf{s} \in \mathcal{S}\})$. According to Lemma C.1, we can infer that:

$$\mathcal{F}^k(\epsilon) = \bigcup_{\{\epsilon_{\mathbf{s}}, \mathbf{s} \in \mathcal{S}\} \in C} \mathcal{F}^k(\{\epsilon_{\mathbf{s}}, \mathbf{s} \in \mathcal{S}\}). \tag{14}$$

Considering that the problems in Definition 3.1 and Definition 3.2 shares the same objective function, we have:

$$\forall \epsilon \ge 0, \quad \exists\{\epsilon_{\mathbf{s}}, \mathbf{s} \in \mathcal{S}\} \in C, \quad \pi_*^{k+1}[\epsilon] \in \mathcal{F}^k(\{\epsilon_{\mathbf{s}}, \mathbf{s} \in \mathcal{S}\}), \tag{15}$$

where $\pi_*^{k+1}[\epsilon]$ is the optimal solution corresponding to the optimal value $J_*^k[\epsilon]$ for the problem in Definition 3.2.

Based on Equation (15), we can derive that

$$\forall \epsilon \ge 0, \quad \exists\{\epsilon_{\mathbf{s}}, \mathbf{s} \in \mathcal{S}\} \in C, \quad \exists \pi \in \mathcal{F}^k(\{\epsilon_{\mathbf{s}}, \mathbf{s} \in \mathcal{S}\}), \quad J_\pi^k[\{\epsilon_{\mathbf{s}}, \mathbf{s} \in \mathcal{S}\}] \ge J_*^k[\epsilon]. \tag{16}$$

Here $J_\pi^k[\{\epsilon_\mathbf{s}, \mathbf{s} \in \mathcal{S}\}]$ is the objective function value in 3.1 with the solution $\pi$ and KL constraints $\{\epsilon_\mathbf{s}, \mathbf{s} \in \mathcal{S}\}$.

Under the KL constraints $\{\epsilon_\mathbf{s}, \mathbf{s} \in \mathcal{S}\}$, the optimal value $J_*^k[\{\epsilon_\mathbf{s}, \mathbf{s} \in \mathcal{S}\}]$ of the problem in Definition 3.1 is no less than $J_\pi^k[\{\epsilon_\mathbf{s}, \mathbf{s} \in \mathcal{S}\}]$. Therefore

$$\exists\{\epsilon_\mathbf{s}, \mathbf{s} \in \mathcal{S}\}, \quad J_*^k[\{\epsilon_\mathbf{s}, \mathbf{s} \in \mathcal{S}\}] \geq J_\pi^k[\{\epsilon_\mathbf{s}, \mathbf{s} \in \mathcal{S}\}] \geq J_*^k[\epsilon]. \tag{17}$$

Proposition 3.4 is proven. **Q.E.D.**

## C.2 Proof of Proposition 3.5

Because the state space $\mathcal{S}$ possibly contains an infinite number of states, the optimization problem in Definition 3.1 is probably a problem with infinite constraints, which is not easy to deal with. Therefore, we first start with a simplified version of the optimization problem, where the state space contains only a finite number of states, and further extend the conclusion on the simplified optimization problem to the original one in Definition 3.1.

The simplified optimization problem is stated as follows:

**Definition C.2** (Simplified optimization problem with point-wise constraints). The simplified optimization problem with point-wise KL constraints on a state space with only a finite number of states $\mathcal{S} = \{\mathbf{s}_1, \mathbf{s}_2, \cdots, \mathbf{s}_m\}$ is defined as

$$\max_\pi \ \mathbb{E}_{\mathbf{s} \sim d_{\pi_\beta}(\cdot), \mathbf{a} \sim \pi(\cdot|\mathbf{s})} \left[Q^{\pi^k}(\mathbf{s}, \mathbf{a}) - V^{\pi^k}(\mathbf{s})\right] \tag{18}$$

$$\text{s.t. } D_{\mathrm{KL}}(\pi(\cdot|\mathbf{s}_i)\|\pi_\beta(\cdot|\mathbf{s}_i)) \leq \epsilon_i, \quad i = 1, 2, \cdots, m \tag{19}$$

$$\int_{\mathbf{a} \in \mathcal{A}} \pi(\mathbf{a}|\mathbf{s}_i)\,\mathrm{d}\mathbf{a} = 1, \quad i = 1, 2, \cdots, m. \tag{20}$$

Here, $\pi^k(k \in \mathbb{N})$ denotes the policy at the $k$-th iteration, $\pi_\beta$ is a behavior policy representing the action selection way in the offline dataset or current replay buffer, and $d_{\pi_\beta}(\mathbf{s})$ is the state distribution of $\pi_\beta$. we utilize the optimal solution for Definition C.2 as $\pi^{k+1}$.

For this simplified optimization problem, we have a lemma below, whose derivation is related to AWR [37] and AWAC [35]:

**Lemma C.3.** *Consider the optimization problem in Definition C.2. The optimal solution of $\pi^{k+1}$, denoted as $\pi_*^{k+1}$, satisfies that $\forall \mathbf{s} \in \{\mathbf{s}_1, \mathbf{s}_2, \cdots, \mathbf{s}_m\}, \mathbf{a} \in \mathcal{A}$,*

$$\pi_*^{k+1}(\mathbf{a}|\mathbf{s}) = \frac{\pi_\beta(\mathbf{a}|\mathbf{s})}{Z_\mathbf{s}} \exp\left(\beta_\mathbf{s}(Q^{\pi^k}(\mathbf{s}, \mathbf{a}) - V^{\pi^k}(\mathbf{s}))\right). \tag{21}$$

*Proof.* The Lagrangian function $L(\pi, \lambda, \mu)$ is given by:

$$L(\pi, \lambda, \mu) = -\mathbb{E}_{\mathbf{s} \sim d_{\pi_\beta}(\cdot), \mathbf{a} \sim \pi(\cdot|\mathbf{s})} \left[Q^{\pi^k}(\mathbf{s}, \mathbf{a}) - V^{\pi^k}(\mathbf{s})\right]$$
$$+ \sum_{i=1}^m \lambda_i \left(\int_\mathbf{a} \pi(\mathbf{a}|\mathbf{s}_i)\,\mathrm{d}\mathbf{a} - 1\right) + \sum_{i=1}^m \mu_i \left(D_{\mathrm{KL}}(\pi(\cdot|\mathbf{s}_i)\|\pi_\beta(\cdot|\mathbf{s}_i)) - \epsilon_i\right), \tag{22}$$

where $\lambda = (\lambda_1, \lambda_2, \cdots, \lambda_m), \mu = (\mu_1, \mu_2, \cdots, \mu_m)$ are Lagrange multipliers. According to the KKT necessary conditions [20], the optimal solution satisfies that $\forall \mathbf{a} \in \mathcal{A}, i \in \{1, 2, \cdots, m\}$,

$$\frac{\partial L}{\partial \pi(\mathbf{a}|\mathbf{s}_i)} = -d_{\pi_\beta}(\mathbf{s}_i)\left(Q^{\pi^k}(\mathbf{s}_i, \mathbf{a}) - V^{\pi^k}(\mathbf{s}_i)\right) + \lambda_i + \mu_i\left(\log\left(\frac{\pi(\mathbf{a}|\mathbf{s}_i)}{\pi_\beta(\mathbf{a}|\mathbf{s}_i)}\right) + 1\right) = 0. \tag{23}$$

Therefore, the optimal policy $\pi_*^{k+1}$ is:

$$\pi_*^{k+1}(\mathbf{a}|\mathbf{s}_i) = \frac{\pi_\beta(\mathbf{a}|\mathbf{s}_i)}{Z_{\mathbf{s}_i}} \exp\left(\beta_{\mathbf{s}_i}(Q^{\pi^k}(\mathbf{s}_i, \mathbf{a}) - V^{\pi^k}(\mathbf{s}_i))\right), \tag{24}$$

where $Z_{\mathbf{s}_i} = \exp\left(\frac{\lambda_i}{\mu_i} + 1\right), \beta_{\mathbf{s}_i} = \frac{d_{\pi_\beta}(\mathbf{s}_i)}{\mu_i}$ are state-dependent factors.

Furthermore, due to the constraint in Equation (20), $Z_{\mathbf{s}_i}$ also equals to $\int_{\mathbf{a} \in \mathcal{A}} \pi_\beta(\mathbf{a}|\mathbf{s}_i) \exp\left(\beta_{\mathbf{s}_i}(Q^{\pi^k}(\mathbf{s}_i, \mathbf{a}) - V^{\pi^k}(\mathbf{s}_i))\right)\,\mathrm{d}\mathbf{a}.$ □

Now we consider extending the conclusion in Lemma C.3 to the more complex optimization problem in Definition 3.1, where the state space $\mathcal{S} = [s_{\min}, s_{\max}]^l$ is a compact set and contains an infinite number of states. Without loss of generality, suppose that $\mathcal{S} = [0, 1]$. The derivation below is easy to transfer to $\mathcal{S} = [s_{\min}, s_{\max}]^l$ with small modifications.

Specifically, we construct a sequence sets $\{B_i, i = 0, 1, 2, \cdots\}$ with each element $B_i = \{j/2^i, j = 0, 1, \cdots, 2^i\}$. We can observe that $\{B_i, i = 0, 1, 2, \cdots\}$ satisfies:

$$\forall i \in \mathbb{N}_+, \quad B_i \subseteq \mathcal{S}; \tag{25}$$

$$\forall i \in \mathbb{N}_+, \quad B_i \subseteq B_{i+1}; \tag{26}$$

$$\forall i \in \mathbb{N}_+, \quad |B_i| = 2^i + 1 < \infty, \quad i.e., \text{ all } B_i \text{ are finite sets, and therefore all } B_i \text{ are compact}; \tag{27}$$

$$\lim_{i \to \infty} \sup_{x \in \mathcal{S}} \inf_{y \in B_i} \|x - y\|_\infty = \lim_{i \to \infty} \frac{1}{2^{i+1}} = 0. \tag{28}$$

The qualities in Equations (25) to (28), together with the assumption in Proposition 3.5 that the feasible space constrained by Equations (2) to (4) is not empty for every $\mathbf{s} \in \mathcal{S}$, meets the prerequisites of the discretization method proposed by [41]. Set $\pi_*^{k+1}[A]$ as the optimal solution of the following optimization problem:

$$\pi_*^{k+1}[A] = \arg\max_\pi \mathbb{E}_{\mathbf{s} \sim d_{\pi_\beta}(\cdot), \mathbf{a} \sim \pi(\cdot|\mathbf{s})} \left[ Q^{\pi^k}(\mathbf{s}, \mathbf{a}) - V^{\pi^k}(\mathbf{s}) \right] \tag{29}$$

$$\text{s.t. } D_{\mathrm{KL}}(\pi(\cdot|\mathbf{s}) \| \pi_\beta(\cdot|\mathbf{s})) \leq \epsilon_\mathbf{s}, \quad \mathbf{s} \in A \tag{30}$$

$$\int_{\mathbf{a} \in \mathcal{A}} \pi(\mathbf{a}|\mathbf{s}) \, d\mathbf{a} = 1, \quad \mathbf{s} \in A, \tag{31}$$

where $A$ is a subset of $\mathcal{S}$. According to the Theorem 2.1 in [41],

$$\pi_*^{k+1}[B_i] \xrightarrow{i \to \infty} \pi_*^{k+1}[\mathcal{S}]. \tag{32}$$

For any $B_i$, because $|B_i| < \infty$, Equation (21) holds for $\pi_*^{k+1}[B_i]$. By combining Equation (21) with Equation (32), we succeed in proving Equation (7) in Proposition 3.5.

Furthermore, we utilize a parameterized policy $\pi_\phi$ to approximate the optimal policy $\pi_*^{k+1}$ in Equation (7), *i.e.*,

$$\phi = \arg\min_\phi \mathbb{E}_{\mathbf{s} \sim d_{\pi_\beta}(\cdot)} \left[ D_{\mathrm{KL}}(\pi_*^{k+1}(\cdot|\mathbf{s}) \| \pi_\phi(\cdot|\mathbf{s})) \right] \tag{33}$$

$$= \arg\max_\phi \mathbb{E}_{\mathbf{s} \sim d_{\pi_\beta}(\cdot)} \left[ \frac{1}{Z_\mathbf{s}} \mathbb{E}_{\mathbf{a} \sim \pi_\beta(\mathbf{a}|\mathbf{s})} \left[ \exp(\beta_\mathbf{s}(Q^{\pi^k}(\mathbf{s}, \mathbf{a}) - V^{\pi^k}(\mathbf{s}))) \log \pi_\phi(\mathbf{a}|\mathbf{s}) \right] \right]. \tag{34}$$

In practice, $Z_\mathbf{s}$ in Equation (34) is challenging to calculate. We follow the derivation of AWR [37] and AWAC [35], and omit the term $\frac{1}{Z_\mathbf{s}}$. Therefore Equation (34) can be rewritten as

$$\phi = \arg\max_\phi \mathbb{E}_{\mathbf{s} \sim d_{\pi_\beta}(\cdot)} \left[ \mathbb{E}_{\mathbf{a} \sim \pi_\beta(\mathbf{a}|\mathbf{s})} \left[ \exp(\beta_\mathbf{s}(Q^{\pi^k}(\mathbf{s}, \mathbf{a}) - V^{\pi^k}(\mathbf{s}))) \log \pi_\phi(\mathbf{a}|\mathbf{s}) \right] \right] \tag{35}$$

$$= \arg\max_\phi \mathbb{E}_{(\mathbf{s}, \mathbf{a}) \sim \mathcal{D}} \left[ \exp(\beta_\mathbf{s}(Q^{\pi^k}(\mathbf{s}, \mathbf{a}) - V^{\pi^k}(\mathbf{s}))) \log \pi_\phi(\mathbf{a}|\mathbf{s}) \right]. \tag{36}$$

Thus Equation (8) in Proposition 3.5 is also proven. **Q.E.D.**

## D  FamO2O's Extension to Non-AWR Algorithms

In this section, we discuss extending FamO2O to non-AWR algorithms, specifically considering the Conservative Q-Learning (CQL) [27].

CQL's policy update rule is given in Equation (37):

$$\pi^{k+1} = \arg\max_\pi \mathbb{E}_{\mathbf{s} \sim \mathcal{D}, \mathbf{a} \sim \pi(\cdot|\mathbf{s})} \left[ \alpha \cdot Q^k(\mathbf{s}, \mathbf{a}) - \log \pi(\mathbf{a}|\mathbf{s}) \right], \tag{37}$$

where $\pi^k$ and $Q^k$ represent the policy and Q function at iteration $k$ respectively, and $\alpha$ is a hyper-paramter. CQL already incorporates a conservative estimation for out-of-distribution data in its Q function update rule, thus lacking a policy constraint for policy updates in Equation (37). However, Equation (37) presents a balance between exploitation (the blue part) and exploration (the red part), determined by $\alpha$. We thus deploy FamO2O to adjust this balance adaptively per state.

Under FamO2O, the policy update rule morphs to Equation (38):

$$\pi_u^{k+1} = \arg\max_{\pi_u} \mathbb{E}_{\mathbf{s}\sim\mathcal{D},\mathbf{a}\sim\pi_u(\cdot|\mathbf{s},\alpha_\mathbf{s})} \left[\alpha_\mathbf{s} \cdot Q^k(\mathbf{s},\mathbf{a}) - \log\pi_u(\mathbf{a}|\mathbf{s},\alpha_\mathbf{s})\right], \tag{38}$$

$$\text{where} \quad \alpha_\mathbf{s} = \pi_b^k(\mathbf{s}), \tag{39}$$

where $\pi_u$ is the universal model taking a state and balance coefficient as input to yield an action, and $\pi_b$ is the balance model mapping a state to a balance coefficient. The changes, denoted in red and blue, depict FamO2O's additional input and the substitution of the adaptive balance coefficient $\alpha_\mathbf{s}$ for the pre-defined one $\alpha$, respectively.

Additionally, the balance model update rule (Equation (40)) aims to maximize the corresponding Q value by finding an optimal $\alpha_\mathbf{s}$, akin to Equation (10).

$$\pi_b^{k+1} = \arg\max_{\pi_b} \mathbb{E}_{\mathbf{s}\sim\mathcal{D}} \left[Q^k(\mathbf{s}, \underbrace{\pi_u^{k+1}(\mathbf{s}, \overbrace{\pi_b(\mathbf{s})}^{\text{balance coefficient } \beta_\mathbf{s}})}_{\text{action}})\right]. \tag{40}$$

As for Q function updates, by consolidating balance model $\pi_b : \mathcal{S} \mapsto \mathcal{B}$ and $\pi_u : \mathcal{S} \times \mathcal{B} \mapsto \mathcal{A}$ into a standard policy $\pi_u(\cdot, \pi_b(\cdot)) : \mathcal{S} \mapsto \mathcal{A}$, these updates remain identical to standard CQL.

# E    Implementation Details

The following section outlines the specifics of baseline (Appendix E.1) and FamO2O (Appendix E.2) implementations.

## E.1    Baseline Implementation

We commence by detailing our baseline implementation. For IQL [24], AWAC [35], and Balanced Replay [28], which have been tested in an offline-to-online RL setting in their respective papers, we utilize the official codes[3,4].

For CQL [27] and TD3+BC [10], we introduce an online fine-tuning process based on the recommended offline RL codes by the authors[5,6]. Specifically, our online fine-tuning for CQL and TD3+BC mirrors the implementation in rlkit, that is, appending samples acquired during online interactions to the offline dataset while maintaining the same training objective during offline pre-training and online fine-tuning.

For SAC [13] and BC, we leverage CQL's codebase, which has already incorporated these two algorithms.

Regarding training steps, we adhere to IQL's experiment setting for a fair comparison across all methods in Figure 5, which involves $10^6$ gradient steps each for offline pre-training and online fine-tuning. It should be noted that due to AWAC's overfitting issue, as discussed by [24], we limit AWAC's offline pre-training to $2.5 \times 10^4$ gradient steps, as recommended. For non-AWR algorithms comparison, namely CQL [27], BR [28], and CQL+FamO2O, we employ $2 \times 10^6$ offline gradient steps, as suggested in CQL's implementation, and $1 \times 10^6$ online gradient steps, aligning with IQL.

---

[3]IQL and AWAC: `https://github.com/rail-berkeley/rlkit`, commit ID c81509d982b4d52a6239e7bfe7d2540e3d3cd986.

[4]Balanced Replay: `https://github.com/shlee94/Off2OnRL`, commit ID 6f298fa

[5]CQL: `https://github.com/young-geng/JaxCQL`, commit ID 80006e1a3161c0a7162295e7002aebb42cb8c5fa.

[6]TD3+BC: `https://github.com/sfujim/TD3_BC`, commit ID 8791ad7d7656cb8396f1b3ac8c37f170b2a2dd5f.

## E.2 Implementation of FamO2O

Next, we provide FamO2O's implementation details. We implement our FamO2O on the official codes of AWAC [35], IQL [24], and CQL [27] discussed above. Our major modification is (i) adding a balance model and (ii) making the original policy conditioned on the balance coefficient (called "universal model" in our paper). Except for these two modifications, we do not change any training procedures and hyperparameters in the original codes.

For the first modification, we employ the stochastic model used by SAC in rlkit to construct our balance model and implement the training process of the balance model described in Section 4.2. For the second modification, we change the input dimension of the original policy from the state dimension $l$ to $l + l_b$, where $l_b$ is the dimension of balance coefficient encodings. We encode balance coefficients in the same way as the positional encoding proposed by [47]. This encoding design expands the dimension of the balance coefficients and avoids the balance coefficients being neglected by the universal model.

For IQL with FamO2O, adaptive balance coefficients are no less than $1.0$ (Locomotion) / $8.0$ (AntMaze) and no larger than $5.0$ (Locomotion) / $14.0$ (AntMaze). For AWAC with FamO2O, the adaptive balance coefficients are no less than $2.0$ (Locomotion) / $9.0$ (AntMaze) and no larger than $3.0$ (Locomotion) / $11.0$ (AntMaze). For CQL with FamO2O, the adaptive balance coefficients are no less than $0.5$ and no larger than $1.5$. To ensure a fair comparison, in Figure 9, the fixed balance coefficient ranges cover the adaptive ranges discussed above.

As for training steps, for IQL with FamO2O, we use $10^6$ gradient steps in the offline pre-training phase and $10^6$ gradient steps in the online fine-tuning phase. For AWAC with FamO2O, we use $7.5 \times 10^4$ gradient steps in the offline pre-training phase and $2 \times 10^5$ (Locomotion) / $1 \times 10^6$ (AntMaze) gradient steps in the online fine-tuning phase. For CQL with FamO2O, we use $2 \times 10^6$ gradient steps in the offline pre-training phase and $1 \times 10^6$ in the online fine-tuning phase.

For more implementation details, please refer to Tables 4 to 6.

## F  More Experimental Results

### F.1  Sensitivity Analysis on Different Ranges of Balance Coefficients

To find out whether FamO2O's performance is sensitive to the range of balance coefficients, we implement FamO2O on IQL [24] with 3 different balance coefficient ranges (*i.e.*, $[1, 5]$, $[1, 4]$, and $[2, 4]$) on 3 datasets of different qualities (*i.e.*, medium-replay, medium, and medium-expert). The results can be found in Figure 12. We further implement FamO2O on IQL with 4 different balance coefficient ranges (*i.e.*, $[6, 12]$, $[8, 12]$, $[8, 14]$, and $[9, 11]$) on 3 datasets of different maze sizes (*i.e.*, umaze, medium, large). The results can be viewed in Figure 13. It can be observed that on various datasets of different qualities or maze sizes, the performance of FamO2O does not vary significantly with different ranges of the balance coefficients. This indicates that within a reasonable scope, FamO2O is insensitive to the range of coefficient balances.

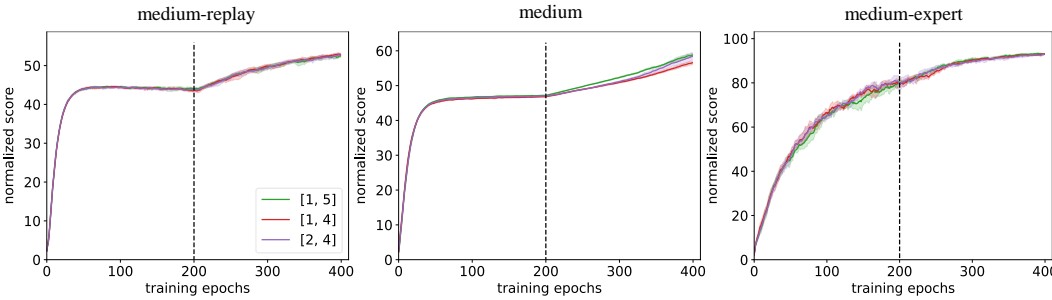

Figure 12: Sensitivity analysis on different ranges of balance coefficients. We choose 3 different ranges (*i.e.*, $[1, 5]$, $[1, 4]$, and $[2, 4]$) and **3 different dataset qualities** (*i.e.*, medium-replay, medium, and medium-expert) in the D4RL [8] HalfCheetah environment. Results are evaluated over 6 random seeds. The shade around curves denotes 95% CIs of the policy performance. Each epoch contains 5k gradient steps. The dashed lines divide the pre-training and fine-tuning.

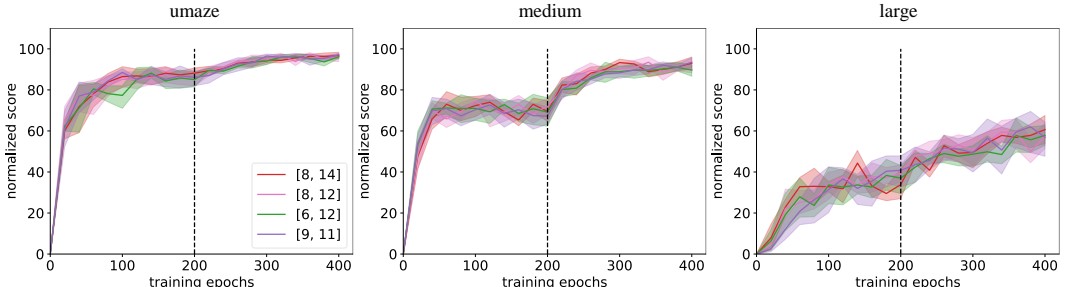

Figure 13: Sensitivity analysis on different ranges of balance coefficients. We choose 4 different ranges (*i.e.*, $[6, 12]$, $[8, 12]$, $[8, 14]$, and $[9, 11]$) and **3 different maze sizes** (*i.e.*, umaze, medium, and large) in the D4RL [8] AntMaze play environments. Results are evaluated over 6 random seeds. The shade around curves denotes 95% CIs of the policy performance. Each epoch contains 5k gradient steps. The dashed lines divide the pre-training and fine-tuning.

### F.2 Comparing FamO2O and Balance Coefficient Annealing

To delve deeper into the efficacy of FamO2O's balance model more thoroughly, we set up a comparison against the balance coefficient annealing. The balance coefficient annealing is a process that gradually augments the balance coefficient value as the fine-tuning stage proceeds. This method is intuitively reasonable, as the initial conservativeness due to the balance coefficient annealing helps to curb extrapolation errors at the commencement of fine-tuning, and as the process continues, this conservativeness is gradually eased to facilitate policy improvement.

We incorporate balance coefficient annealing into IQL [24], named IQL+annealing, providing a basis for comparison with IQL+FamO2O. The comparison results are presented in Table 3. The results show that FamO2O exceeds the performance of balance coefficient annealing across most of the D4RL Locomotion datasets [8].

Table 3: Comparing FamO2O and balance coefficient annealing, FamO2O notably surpasses the latter on the majority of D4RL locomotion datasets [8], demonstrating statistically significant performance improvement.

| Dataset | FamO2O | Annealing |
|---|---|---|
| hopper-mr-v2 | 97.64 | 87.82 |
| hopper-m-v2 | 90.65 | 72.98 |
| hopper-me-v2 | 87.28 | 72.26 |
| halfcheetah-mr-v2 | 53.07 | 53.35 |
| halfcheetah-m-v2 | 59.15 | 57.37 |
| halfcheetah-me-v2 | 93.10 | 93.91 |
| walker2d-mr-v2 | 92.85 | 87.29 |
| walker2d-m-v2 | 85.50 | 85.89 |
| walker2d-me-v2 | 112.72 | 112.72 |
| **total** | **771.96** | **723.61** |
| **95% CIs** | **753.52~788.51** | **704.21~744.43** |

In addition, we calculate the 95% CIs for both IQL+FamO2O and IQL+annealing in Table 3, following the suggestion by rliable [3]. The lower CI for IQL+FamO2O is mush higher than the upper CI for IQL+annealing, signifying a statistically significant improvement of FamO2O over balance coefficient annealing. These findings establish that in comparison to annealing, FamO2O showcases a more sophisticated and effective approach in adaptively modulating the balance between improvement and constraint.

## G Potential Negative Societal Impacts

This work proposes a plug-in framework, named FamO2O, which can be implemented on top of the existing offline-to-online RL algorithms to further improve policy performance. Therefore, the potential negative societal impacts of our method are similar to those of the offline-to-online RL. Generally, FamO2O would greatly improve policy performance. However, as with most RL algorithms, FamO2O cannot guarantee to take safe and effective actions in all kinds of scenarios. We advocate that the users of FamO2O should be aware of the potential consequences and utilize FamO2O safely, especially in online environments such as self-driving, robotics, and healthcare.

Table 4: Details of FamO2O implemented on **IQL** [24] for the **D4RL Locomotion and AntMaze** benchmark [8]. Except for our newly added balance model, the hyperparameters of our method keep the same as those of the vanilla IQL.

| | Name | Value |
|---|---|---|
| Balance model | optimizer | Adam [21] |
| | learning rate | $3 \times 10^{-4}$ |
| | update frequency | 5 (Locomotion) / 1 (AntMaze) |
| | hidden units | $[256, 256]$ |
| | activation | ReLU [1] |
| Universal model | optimizer | Adam [21] |
| | learning rate | $3 \times 10^{-4}$ |
| | update frequency | 1 |
| | hidden units | $[256, 256]$ |
| | activation | ReLU [1] |
| $Q$ function model | optimizer | Adam [21] |
| | learning rate | $3 \times 10^{-4}$ |
| | update frequency | 1 |
| | hidden units | $[256, 256]$ |
| | activation | ReLU [1] |
| | target $Q$ soft update rate | $5 \times 10^{-3}$ |
| $V$ function model | optimizer | Adam [21] |
| | learning rate | $3 \times 10^{-4}$ |
| | update frequency | 1 |
| | hidden units | $[256, 256]$ |
| | activation | ReLU [1] |
| | target $V$ soft update rate | $5 \times 10^{-3}$ |
| | quantile | 0.7 (Locomotion) / 0.9 (AntMaze) |
| Other training parameters | batch size | 256 |
| | replay buffer size | $2 \times 10^6$ |
| | discount | 0.99 |

Table 5: Details of FamO2O implemented on **AWAC** [35] for the **D4RL Locomotion and AntMaze** benchmark [8]. Except for our newly added balance model, the hyperparameters of our method keep the same as those of the vanilla AWAC.

|  | Name | Value |
|---|---|---|
| Balance model | optimizer | Adam [21] |
|  | learning rate | $3 \times 10^{-4}$ |
|  | update frequency | 5 |
|  | hidden units | $[256, 256]$ |
|  | activation | ReLU [1] |
| Universal model | optimizer | Adam [21] |
|  | learning rate | $3 \times 10^{-4}$ |
|  | update frequency | 1 |
|  | hidden units | $[256, 256, 256, 256]$ |
|  | activation | ReLU [1] |
|  | weight decay | $1 \times 10^4$ |
| $Q$ function model | optimizer | Adam [21] |
|  | learning rate | $3 \times 10^{-4}$ |
|  | update frequency | 1 |
|  | hidden units | $[256, 256]$ |
|  | activation | ReLU [1] |
|  | target $Q$ soft update rate | $5 \times 10^{-3}$ |
| Other training parameters | batch size | 1024 |
|  | replay buffer size | $2 \times 10^6$ |
|  | discount | 0.99 |

Table 6: Details of FamO2O implemented on **CQL** [27] for the **D4RL Locomotion** benchmark [8]. Except for our newly added balance model, the hyperparameters of our method keep the same as those of the vanilla AWAC.

|  | Name | Value |
|---|---|---|
| Balance model | optimizer | Adam [21] |
|  | learning rate | $3 \times 10^{-4}$ |
|  | update frequency | 1 |
|  | hidden units | $[256, 256]$ |
|  | activation | ReLU [1] |
| Universal model | optimizer | Adam [21] |
|  | learning rate | $3 \times 10^{-4}$ |
|  | update frequency | 1 |
|  | hidden units | $[256, 256]$ |
|  | activation | ReLU [1] |
| $Q$ function model | optimizer | Adam [21] |
|  | learning rate | $3 \times 10^{-4}$ |
|  | update frequency | 1 |
|  | hidden units | $[256, 256]$ |
|  | activation | ReLU [1] |
|  | target $Q$ soft update rate | $5 \times 10^{-3}$ |
| Other training parameters | batch size | 256 |
|  | replay buffer size | $2 \times 10^6$ |
|  | discount | 0.99 |

