# OpenReview forum: "Train Once, Get a Family: State-Adaptive Balances for Offline-to-Online Reinforcement Learning"
_NeurIPS.cc/2023/Conference — NeurIPS 2023 spotlight_

### Official Review · Reviewer_BkWu · 2023-06-18

**Soundness:** 3 good
**Presentation:** 3 good
**Contribution:** 3 good
**Rating:** 7
**Confidence:** 3

**Summary:**

Authors propose a framework for offline-to-online tuning of offline RL algorithms tunning. The idea is to train an additional network which desides helps to keep improvement-constraint balance during finetuning.

**Strengths:**

Approach improves all of the checked algorithms performance and can be applied to different offline RL algorithms. Good range of benchmarking tasks and  algorithms.

**Weaknesses:**

-

**Questions:**

How does the modification affect the compute time required to  train algorithms?

**Limitations:**

Adding another network requires additional choice of  hyperparameters which might be hard to find

---

> ### Author Rebuttal · Authors · 2023-08-08
>
> Your keen observations and valuable suggestions are highly appreciated, and we thank you for helping us to strengthen our paper.
>
> > Q1: How does the modification affect the compute time required to train algorithms?
>
> Thank you for raising this question. To examine the computational overhead introduced by FamO2O, we utilized IQL and FamO2O+IQL as examples, assessing their training time using their JAX implementations over 1M offline steps and 1M online steps. The result is provided in the table below. From this analysis, we can discern that FamO2O augments the training time of IQL by roughly 8 minutes. **Given the substantial performance improvement brought by FamO2O, this increase in training time can be deemed acceptable.**
>
> | (unit: minute)             | IQL    | FamO2O+IQL |
> | -------------------------- | ------ | ---------- |
> | offline pre-training phase | 9.278  | 11.48      |
> | online fine-tuning phase   | 12.532 | 18.25      |
> | total                      | 21.81  | 29.73      |
>
> > Q2: Adding another network requires additional choice of hyperparameters which might be hard to find.
>
> Thank you for your insightful observation. We do recognize the complexity that adding another network introduces due to the need to choose additional hyperparameters, such as the selection of the space of balance coefficients $\mathcal{B}$. Nevertheless, our empirical results, as presented in Figure 11 and Figure 12 of our paper, demonstrate that **FamO2O's performance is largely unaffected by the choice of hyperparameters, e.g., $\mathcal{B}$, within a sensible range**.
>
> ---
>
> We extend our heartfelt gratitude once more for your thorough review and thoughtful comments. We anticipate further dialogue and collaboration, and are open to any more thoughts you may have to help refine our work.

---

> > ### Comment · Reviewer_BkWu · 2023-08-11
> >
> > Thank you for answering my questions. For hyperparameters sensitivity demonstration you can consider using EOP https://arxiv.org/abs/2110.04156. I have no further questions on your work and will increase my confidence score.

---

> > > ### Author Response · Authors · 2023-08-15
> > > **Thank you**
> > >
> > > Thank you for your thoughtful comments and advice. We have carefully examined the EOP paper and will take it into consideration for inclusion in the next version of our work. Your insights are greatly appreciated.

---

### Official Review · Reviewer_urL3 · 2023-06-20

**Soundness:** 3 good
**Presentation:** 3 good
**Contribution:** 3 good
**Rating:** 7
**Confidence:** 4

**Summary:**

This paper approaches offline-to-online RL from the intuition that at a particular state, if the dataset already contains good actions, then the subsequent online tuning should be more conservative to retain the good actions in the dataset; but if the dataset's actions is poor, then more radical policy improvement is needed.
To this end, this paper introduces a framework, Family Offline-to-Online RL (FamO2O), which aims at a state-adaptive improvement-constraint balance for each state.
Specifically, from the collected dataset, the authors train a diverse policy family ranging from conservative to radical and use the environmental feedback to select an appropriate policy from this family at each state.
Practically, this is achieved by a universal model, which determines the degree of policy conservatism; and a balance model, which learns the balance coefficients at each state.
Experimental results show that FamO2O improves on various offline-to-online RL methods and achieves competitive performance.

**Strengths:**

1. The paper is well-written and generally easy-to-follow.
2. The proposed method is well theoretically justified.
3. The empirical discussion and ablation study are thorough.

**Weaknesses:**

1. The proposed method seems require abundant diverse data, which may not be feasible on harder settings, e.g., the Adroit domain in the D4RL benchmark where the data is limited and the data distribution is narrow and lack of diversity.
2. Maybe I misunderstand something. I think the purpose of the $\pi_b(s)$ model is to find the $\beta_s$ that corresponds to the optimal sequence of constraints $\\{\epsilon_s, s \in \mathcal{S}\\}$  in Eqn. (6), which relates to $\beta_s$ by the Language multipliers $\mu(s)$ (Line 537, Appendix C.2). In this regard, the paper's story of balancing policy improvement and constraint can be slightly over-complicated and somewhat confusing.
3. The proposed method is in spirit similar to decision-transformer-style methods, and is therefore less a surprise.


**Questions:**

1. Could you explain more on how the updating rule Eqn. (10) train the balance model $\pi_b$ to control policy improvement and constraint? Why does maximizing the $Q^k$ values achieve this balance?
2. The significant variation in data quality across different states depicted by Figure 2 may come from the nature of the medium-expert datasets, which by construction contains medium and expert trajectories. Does this variation in data quality exist in other type of datasets, say expert, medium, or medium replay? More generally, is this phenomenon of significant data-quality variation ubiquitous or specific?
3. Could you explain more on the benefit of "state-adaptive improvement-constraint balances"? Even if we do not make such a balance and simply just online finetune an offline-pretrained policy, the environmental feedback should be able to tell us which actions are good and should be retained and which actions are poor and should be improved.
4. Could you explain more why you consider the term $\log \pi (a|s)$ in Eqn. (1) as a policy constraint? What is the target of this constraint, especially when $(s,a)$ is an online interaction samples?
5. Could you explain more on how you get Eqn. (12) and (13) from Eqn. (5)? It would be better if you can expand Appendix C.1 to include more details and explanations. The current version is a bit hard to follow.
6. nit: in Eqn. (6), are you missing a "$\forall \epsilon > 0$" before $\exists \\{...\\}$?
7. [L161] Could you explain more on the cooperation between $\pi_u$ and $\pi_b$? Why couldn't we still randomly sample $\beta_s$ during online fine-tuning?
8. [L159] How many $\beta_s$ vectors are required to learn the universal model? How does this number scale with the number of states in the dataset? And how to select/design the balance coefficient space $\mathcal{B}$?

**Limitations:**

The authors adequately addressed the limitations.

---

> ### Author Rebuttal · Authors · 2023-08-07
>
> We want to express our thanks to you for the detailed feedback and constructive criticism that guided our revisions.
>
> > Q1: The proposed method seems to require abundant diverse data.
>
> Thanks for pointing this out. The confusion might stem from an unclear explanation in our manuscript, and we'll clarify it in the paper's next version:
>
> It's worth noting that **our method doesn't necessarily need varied data qualities**. We claim that it can determine the best conservative/radical balance for each state in online scenarios based on the data quality in the collected dataset. **If the dataset is diverse in quality, the balances will be diverse; if the quality is consistent, the balances will be correspondingly consistent** (please refer to Figure 3 in [the global rebuttal's attached PDF](https://openreview.net/attachment?id=1S5GhI6UFd&name=pdf) for empirical evidence). Even with consistent data quality, our method has **the advantage of adaptively finding the proper balance** over existing offline RL algorithms with human-chosen balances.
>
> ---
> > Q2: The paper's story of balancing policy improvement and constraint can be slightly over-complicated and somewhat confusing.
>
> Thank you for your question, and apologies for any confusion. Here's the logic behind our method:
> - Our method aims to balance policy improvement and constraint, equating to different upper bounds $\epsilon_{\mathbf{s}}$ (see Equation 3).
> - Unlike AWAC or IQL, which use a single constraint term (Equation 5), different $\epsilon_{\mathbf{s}}$ create $|\mathcal{S}|$ constraints (Equation 4), requiring $|\mathcal{S}|$ Lagrange multipliers.
> - Given $\beta = d_{\pi_\beta}(\mathbf{s})/\mu$ (Line 537), we must have $|\mathcal{S}|$ different $\beta$ values, hence the necessity for state-adaptive balance (Proposition 3.5).
> - To generate these $\beta$ values, we introduce $\pi_b: \mathcal{S}\mapsto \mathbb{R}$, outputting $\beta$ based on $\mathbf{s}$.
>
> We will add this explanation to the next version of our paper.
>
> ---
> > Q3: The proposed method is in spirit similar to decision-transformer-style methods.
>
> Thank you for pointing out the similarity. **Please refer to [our response to Reviewer APCr's Q3](https://openreview.net/forum?id=vtoY8qJjTR&noteId=EQ1sNpef2S)**, where we compare our method with other works **including decision transformer [9]**.
>
> ---
> > Q4: How does the updating rule Eqn. (10) train the balance model $\pi_b$ to control policy improvement and constraint? Why does maximizing the $Q^k$ values achieve this balance?
>
> Thank you for your question. In Equation (9), the universal model $\pi_u$ is trained to adjust the conservative/radical balance of the policy based on the inputted balance coefficient $\beta_s$. Building on this, Equation (10) trains the balance model to select the $\beta_s$ that enables $\pi_u$ to maximize the Q value. This approach is grounded in the understanding that **the Q value serves as an estimate of future return, which is our ultimate goal of striking a balance between policy improvement and constraint**. This is in alignment with Equation (2), where it's important to note that $V(s)$ has no gradients concerning actions, and consequently, no gradients with respect to $\pi_b$. We will include this clarification in the next version of our paper.
>
> ---
> > Q5: Is the phenomenon of data-quality variation ubiquitous or specific?
>
> As depicted in **Figure 1 of [the global rebuttal's PDF](https://openreview.net/attachment?id=1S5GhI6UFd&name=pdf)**, the phenomenon of substantial variation in data quality is a **widespread occurrence** in offline datasets. It's worth reiterating, as previously noted in response to your Q1, that while our method can harness data across a range of qualities, **it does not fundamentally depend on data quality diversity**.
>
> ---
> > Q6: Could you explain more on the benefit of "state-adaptive improvement-constraint balances"?
>
> Indeed, as you suggest, online feedback can help the agent determine which actions are good and which are bad even without our state-adaptive improvement-constraint balances. However, **the existing fixed balance methods present two primary problems, which can be addressed by the implementation of state-adaptive balances**:
>
> 1. **Difficulty in Deciding Proper Balance:** In the absence of abundant information about an offline dataset and online environment, during offline pre-training, it's **difficult to pre-specify a proper improvement-constraint balance** that can optimally deal with every future state encountered during online interaction.
>
> 2. **Inflexibility of Fixed Balances:** As indicated by [3], existing offline-to-online algorithms with fixed balances inhibit drastic changes in an agent's behavior (e.g., from conservative to radical, or vice versa) due to "primacy bias." In contrast, our approach facilitates flexible adjustment of conservative or radical degrees for different states during online inference, as stated in our paper. **Without our proposed state-adaptive balances, the algorithms would struggle to adapt the conservative/radical degrees appropriately for varying states**.
>
> ---
> **Because of the character limit and our desire to respond to your questions thoughtfully, we've placed the remaining rebuttal in the [global rebuttal](https://openreview.net/forum?id=vtoY8qJjTR&noteId=1S5GhI6UFd). Please refer to it for the follow-up.**
>
> ---
>
> Again, we thank you for your invaluable insights and support in enhancing our work. We are eager to engage in further discussions or address any additional concerns to continue improving our manuscript.

---

> > ### Comment · Reviewer_urL3 · 2023-08-16
> > **Response to the authors**
> >
> > Dear authors,
> >
> > Thank you so much for the detailed response, which clears out all my questions. I will increase my rating to 7.

---

> > > ### Author Response · Authors · 2023-08-16
> > > **Thank you**
> > >
> > > Dear Reviewer urL3,
> > >
> > > Thank you for your thoughtful review and for taking the time to reconsider our work. We appreciate your positive feedback and the increased rating. Your insights have been invaluable to us, and we are pleased to have addressed your concerns.
> > >
> > > Best regards.

---

### Official Review · Reviewer_APCr · 2023-06-21

**Soundness:** 4 excellent
**Presentation:** 4 excellent
**Contribution:** 3 good
**Rating:** 7
**Confidence:** 4

**Summary:**

The paper proposes a new algorithm to perform offline-to-online reinforcement learning. The core idea is to consider a state-adaptive balance parameter, which aims to encourage imitation of dataset behavior only if the corresponding advantages / values are high, while prior works have mostly assumed fixed balances. The authors provide detailed experimental evaluation of their approach and show superiority over relevant baselines.

**Strengths:**

I think offline-to-online RL is a very relevant and promising direction for research since if reliable it would enable much more practical applications of RL in real-world tasks. The core idea of the paper, training a collection of policies that adapts on a per state level, appears logical and powerful (however please see [2,3,4], which I think put forward similar ideas) and will in the future probably significantly influence the way offline-to-online RL is performed and thought about. The paper is overall very well written and understandable & offers a strong statistical analysis of the proposed algorithms performance compared to relevant baselines.

**Weaknesses:**

The term data quality is frequently used but not really introduced (could mean accuracy / truth of the information in the data, but I think it means something like return).

Not all D4RL locomotion datasets were used - its unclear whether that's an issue since the selection is not justified.

The considered baselines appear relevant (TD3+BC & CQL less so since its already shown in the AWAC paper that using the same conservative formulations in the online setting does not work well), however I think the closest existing methods are missing: E.g. Confidence-conditioned value functions [1] automatically adjusts its policy confidence level (as far as I understand on a per state level since every state goes into the considered history). Also methods like RvS [4], which condition on return, could easily be extended to this setting (i.e. always try to maximise conditioned return during online data collection). Generally I have the feeling that some very relevant related work regarding offline adaptive policies was not considered, i.e. also:
[2] User-Interactive Offline RL (which considers policies conditioned on a balance between conservative and very liberal)
[3] offline policies should be trained to be adaptive (which adapts policies based on the online collected history)
I'm not sure the state-adaptive balance between conservative and radical idea for offline-to-online is thus entirely novel.

I see some issues regarding clarity that could easily be fixed (see questions)

**Questions:**

Could you please define how you use the term data quality (I'm assuming it has something to do with the data's return, but I'm not sure)?

Please elaborate why the random datasets in the D4RL datasets were not considered.

There exist offline-to-online methods that already consider something like state-adaptive policy changes, like confidence-conditioned value functions [1], as well as other adaptive offline-to-online policy concepts [2,3] that could easily be extended to your case - why were they not considered? (I realise that some are rather recent, but I think they deserve at least brief discussion)

I do not understand section 6.1 / figure 8 at all - what is meant by guidance? what is meant by high / low quality data? Starting positions are denoted as triangles, suggesting a moving direction but I don't think that's intended. The text says high balance coefficients are found where data quality is also high (quality=reward / return?) & that higher quality data is found at the lower crossing point in the 5th row, however from the color encoding it seems that the balance coefficient there is actually the lowest...

I think the ratio between offline and online training steps as well as collected interaction steps is a crucial parameter for reproducibility / future comparisons, however that information is only conveyed in the figure 10 axis & the appendix - I think it should be explicitly stated somewhere in the experimental section text.

When I look at figure 10, it seems that after all online training steps, the attained performance is almost identical to that at the end of offline training - isn't that a very disappointing result and couldn't you just throw away the whole online part altogether then? Since FamO2O outperforms prior offline-to-online baselines, does that mean that these baselines only get worse with online training?

[1] Hong, J., Kumar, A., & Levine, S. (2022). Confidence-Conditioned Value Functions for Offline Reinforcement Learning. ICLR 2023
[2] Swazinna, P., Udluft, S., & Runkler, T. (2022). User-Interactive Offline Reinforcement Learning. ICLR 2023
[3] Ghosh, D., Ajay, A., Agrawal, P., & Levine, S. (2022, June). Offline rl policies should be trained to be adaptive. ICML 2022
[4] Emmons, S., Eysenbach, B., Kostrikov, I., & Levine, S. (2021). RvS: What is Essential for Offline RL via Supervised Learning?. arXiv preprint arXiv:2112.10751.

**Limitations:**

no, limitations are not discussed - perhaps looking at the not improved performance after online fine-tuning in figure 10, as well as the often only small improvements over standard IQL, a brief discussion would be good.

---

> ### Author Rebuttal · Authors · 2023-08-07
>
> Your thoughtful comments and critiques are sincerely appreciated and have been instrumental in refining our study. **Due to the character limit, the references in this rebuttal are at the global rebuttal.** Thanks for your time and effort.
>
> > Q1: Definition of the term data quality?
>
> Thank you for pointing this out. Your assumption that **the "data quality" refers to the return** is correct. Although we have indeed implicitly mentioned this in Figure 2 and Lines 44-48 of our manuscript, we will make sure to explicitly define the term in the next version of our paper.
>
> ---
>
> > Q2: Elaborate on why the random datasets in the D4RL datasets were not considered.
>
> Thank you for your question. Our approach mainly **follows the practice in IQL paper**, and many other works (e.g., [1,2]) also exclude tests on random datasets. **The primary reason is the RL agent's primacy bias** [3], leading to overfitting on poor-quality random datasets, resulting in performance that often does not exceed that of directly training online for the same number of gradient steps. To confirm this, we tested IQL with offline-to-online learning on `hopper-random-v2` and compared it with SAC's online learning performance. The results, as shown in the table below, demonstrate that IQL's performance on random datasets is significantly lower than SAC's. Hence, using random datasets for offline-to-online RL is considered to be of limited practical value in our context.
> |                                                            | Normalized Score Mean | Normalized Score Std |
> | ---------------------------------------------------------- | --------------------- | -------------------- |
> | IQL (offline-to-online, pre-trained on `hopper-random-v2`) | 40.02667              | 38.79444             |
> | SAC (online learning)                                      | 86.86755              | 21.07426             |
>
> ---
> > Q3: Other related works [5,6,7,8]
>
> Thank you for pointing out these related works. Our FamO2O method emphasizes two main features: 1) Utilizing various data, similar to Balanced Replay [4], and 2) employing a conditioned policy, related to the papers mentioned by you [5,6,7,8] and by Reviewer urL3 [9]. Having discussed Balanced Replay extensively, we will now focus on the conditioned policy's related works:
>
> 1. **[5]:** They focus on diverse conservative **Q-values**, while we prioritize varying conservative **policies**. Their method **isn't suited for continuous action settings** like D4RL, but ours is versatile for both continuous and discrete actions.
> 2. **[6]:** Their setting relies on **user interaction** for online adjustments, whereas ours is **automatic**.
> 3. **[7]:** This paper emphasizes offline RL, estimating probabilities of various MDPs in uncovered areas with Bayesian posterior, and adjusting policies accordingly. However, in the offline-to-online RL context, where agents explore during online fine-tuning, **uncoverage may not be a primary concern**. Their focus is on **adapting to different MDPs**, while ours targets **utilizing data of varying quality**.
> 4. **[8, 9]:** Despite sharing conditioned policy use (as discussed in our related work section), they aim to liberalize policy learning from merely copying offline behavior, **differing significantly from our motivation**, and their strategies are **not adaptively adjustable**.
>
> We value your insight and will include this discussion in the related work section.
>
> ---
> > Q4: Queries on guidance, high/low-quality data, starting positions & triangles, and texts in L258-262
>
> Thank you for your comments and questions regarding Section 6.1 and Figure 8. We apologize for any confusion caused and provide explanations here:
>
> 1. **Guidance**: Guidance refers to directing the agent to the shortest route across a crossing point during offline data collection. Without it, the agent moves randomly.
> 2. **High/Low-Quality Data**: High-quality data results from the agent moving w/ guidance, while low-quality data is collected when the agent moves w/o guidance.
> 3. **Starting Positions & Triangles**: The triangles don't represent moving direction. We've revised the figure to avoid confusion (see Figure 2 in the global rebuttal's PDF).
> 4. **Texts in L258-262**: There were typos in lines L258-262. The corrected statement is: "Figure 8(b) shows the agent typically outputs **lower** balance coefficients for **high-quality** samples and **higher** ones for **low-quality** data." This aligns with the FamO2O motivation, as lower/higher balances reflect a more conservative/radical policy.
>
> We will add the above clarifications and modifications to the next version.
>
> ---
> > Q5: Explicitly state crucial parameters in the experimental section.
>
> Thanks for your constructive suggestions. We will state the crucial parameters, e.g., offline and online training steps, and collected interaction steps, in the experimental section in the next version.
>
> ---
> > Q6: The almost identical performances after online fine-finetuning and offline pre-training in Figure 10.
>
> Thank you for your question. Figure 10 specifically highlights **an extreme case** on a single dataset, where IQL shows **the most significant performance drop**. It demonstrates that even in this situation, IQL+FamO2O can alleviate the drop and attain good performance. However, this is an isolated instance, and generally, **as shown in the table below, both IQL and IQL+FamO2O achieve better performances after online fine-tuning compared to after offline pre-training**.
>
> |            | Offline Performance Sum | Online Performance Sum | Fine-tuning Improvement |
> | ---------- | ---------- | ------ | ----------------------- |
> | IQL        | 581.7| 718.3| +136.6|
> | FamO2O+IQL | 584.4| 772.0| +187.6|
>
> ---
> > Q7: Limitations are not discussed.
>
> Thanks for your advice. We will discuss limitations in the revised version.
>
> ---
> Thank you once again for your valuable feedback and advice. We look forward to further discussion to refine our work.

---

> > ### Comment · Reviewer_APCr · 2023-08-14
> > **Rebuttal Response**
> >
> > Thank you very much for the detailed responses to my questions, they have been very helpful in better understanding your work, especially Q3,4 & 6.
> >
> > I am however still not quite sure I understand Figure 8 (Q4): When you say guidance means "directing the agent", how exactly do you direct it? Is there an explicit reward signal given only at this point or does a separate policy take over which "knows" the way or ...? I believe illustrative examples like this one are important & I understand you have limited space, but I think a little more information is needed to make it really helpful.
> >
> > If the color encoding is correct & what you wrote in your response
> > > the agent typically outputs lower balance coefficients for high-quality samples
> >
> > is correct, I might still misunderstand your method. In Eq(1) it seems to me that high balance coefficients would lead to the agent more likely copying the behaviour that was present in the dataset - since I would like to repeat behaviour that has yielded high return, I would expect high balance coefficients in states where you have high quality data. In the example it is however the other way around... Could you please elaborate?
> >
> > One more clarification regarding your answer on Q6: Does that mean the plot shows performance only on a single offline dataset? If so,  which one?

---

> > > ### Author Response · Authors · 2023-08-15
> > > **Explanations of Figure 8 (Q4), Eq. (1), and Figure 10 (Q6)**
> > >
> > > Thank you for your thought-provoking questions. We've addressed your inquiries in detail below.
> > >
> > > > The meaning of "directing the agent".
> > >
> > > "Directing the agent" refers to compelling the agent to adhere to the route and direction that yield the shortest path to the goal, rather than letting the agent decide the route and direction on its own. This aligns with your perception that "a separate policy take over which 'knows' the way".
> > >
> > > Thank you for bringing this to our attention. We will incorporate the above explanation into the next version of our paper to make this point clearer.
> > >
> > > > The effect of the balance coefficient value.
> > >
> > > Thank you for your thoughtful question. You're suggesting that high balance coefficients should be used with high-quality data, which makes sense at first glance. But our method works differently, and here's how:
> > >
> > > - **For High-Quality Data:** Utilizing high balance coefficients might lead the policy to aggressively pursue actions with the highest possible advantage, $Q(\mathbf{s}, \mathbf{a})-V(\mathbf{s})$. **But since the advantages of the high-quality data are already high, trying to push for even higher advantages can easily lead to mistakes due to overestimation in Q values**. So, we use lower balance coefficients for high-quality data, making sure the policy stays safe by following the known good actions.
> > > - **For Low-Quality Data:** On the other hand, with low-quality actions, it makes sense to use higher balance coefficients. **Copying what the low-quality data does will surely end up in failure, so it's worth the risk to try for something better**. This leads to a more daring or "radical" policy that looks for higher-quality actions.
> > >
> > > By using balance coefficients this way, depending on whether the data is high or low quality, our method reduces the risks and finds a good middle ground. It doesn't chase after the highest advantages in a way that can cause mistakes, but it also doesn't just copy what's in the bad data. It's a careful balance that helps the policy make the best decisions.
> > >
> > > > One more clarification regarding your answer on Q6: Does that mean the plot shows performance only on a single offline dataset? If so, which one?
> > >
> > > Yes, Figure 10 displays the performance solely on one offline dataset, namely `antmaze-umaze-diverse`, as referenced on line L297 of our manuscript. We selected this specific dataset because, on it, IQL exhibits the most significant decline in performance when transitioning from offline pre-training to online fine-tuning. Although we've alluded to this rationale on lines L296-297 of our manuscript, we will articulate it more explicitly in the upcoming version of our paper.

---

> > > > ### Comment · Reviewer_APCr · 2023-08-17
> > > > **Rebuttal Response 2**
> > > >
> > > > Thank you for the additional clarifications. From your response regarding the balance coefficients, I am however now not sure whether I understand the balance model correctly:
> > > >
> > > > You say that low balance coefficients should be provided by $\pi_b$ if the data quality (i.e. return) in the current state is expected to be high. However, as far as I understand the balance model training, it basically just maximises Q-value (i.e. expected return) of the resulting universal model actions. If what you say is correct, then that means that the Q-maximizing actions are provided by the universal model when the input balance coefficient is low - this is possible, but I wonder why it would be the case - is there any extra effect / regularisation that prefers low balance coefficients if quality / Q-value is high?
> > > >
> > > > I'll try to give a simplified example: If I optimise Eq. 9 not over the entire dataset, but only a single state, and this state has high quality data, then the value in the expectation is maximised if high balance coefficients coincide with high log likelihoods from the universal policy model. Of course the universal model could output the highest log likelihood for low balances, but why would it / what mechanism is pulling the likelihood maximising actions towards the low balance coefficients in the universal model?

---

> > > > > ### Author Response · Authors · 2023-08-17
> > > > > **Explanation of the Balance Coefficient Selection Mechanism**
> > > > >
> > > > > Thank you for your thoughtful question. It brings attention to a critical aspect of our methodology that may appear perplexing at first glance.
> > > > >
> > > > > Indeed, Equation (9) does not seem to explicitly contain any effect or regularization to guide the balance model, $\pi_b$, to choose low or high balance coefficients, $\beta$, depending on data quality. **However, there is an implicit effect -- namely, online rewards -- that directs the behavior of the balance model in the desired manner.**
> > > > >
> > > > > To elucidate this mechanism, let's focus on states with high-quality data (the process for low-quality data is the inverse). During **the initial stage of online fine-tuning**, selecting a large $\beta$ for high-quality data states often leads to an action with an overestimated Q-value. This, in turn, results in comparatively low online rewards, subsequently causing lower Q-values. Thus, $\pi_b$ learns to shy away from large $\beta$ values for high-quality data states. **As online fine-tuning proceeds**, because small $\beta$ values are selected more often for high-quality data states, the universal policy $\pi_u$ becomes increasingly trained with small $\beta$, enhancing its performance compared to large $\beta$ values.
> > > > >
> > > > > **Returning to your simplified example** involving a single state with high-quality data, the mechanism described above still holds. The online rewards will lead the balance model $\pi_b$ to favor smaller balance coefficients $\beta$ for the universal model $\pi_u$. **Therefore, even in this minimalist scenario, the interplay between the balance coefficients and online rewards enforces the selection strategy we have described.**

---

> > > > > > ### Comment · Reviewer_APCr · 2023-08-18
> > > > > > **Rebuttal Response 3**
> > > > > >
> > > > > > Thank you for your insightful explanation.
> > > > > >
> > > > > > I had considered the setting more from an offline perspective at first, where my intuition was high quality should lead to large β. However, I hadn't thought about the fact that this leads to low returns in the online finetuning due to overestimation & that the online feedback would then turn the whole thing around since the balance model needs to counteract. Thank you for clearing up the confusion.
> > > > > >
> > > > > > While thinking about this, I came up with another question:
> > > > > > How do you determine the balance coefficient ranges and how do you enforce them?
> > > > > >
> > > > > > I.e. it seems from appendix fig.11 & 12 that the algorithm is not particularly sensitive the chosen interval, yet different ones are used for the different environments - why is that? I assume it has something to do with the reward scale? How would I choose the interval when I would deploy FAMO2O in a new environment? Lastly, do you clip network outputs to the relevant range or how do you enforce valid β values?

---

> > > > > > > ### Author Response · Authors · 2023-08-18
> > > > > > > **Reply to Rebuttal Response 3**
> > > > > > >
> > > > > > > Thank you for your insightful inquiry. We're glad that our previous explanations were clear, and we welcome the opportunity to address your new questions.
> > > > > > >
> > > > > > > ---
> > > > > > >
> > > > > > > > Choosing the range of $\beta$ when applying FamO2O to a new algorithm
> > > > > > >
> > > > > > > In selecting the range of $\beta$ for FamO2O, we adhere to **two principles** that have guided our work:
> > > > > > > (1) The region should encompass the original algorithm's $\beta$, as this value has been carefully chosen for the algorithm, providing an indicator for the general scale of the range;
> > > > > > > (2) The upper and lower bounds of the $\beta$ range should lead to reasonable performance in the original algorithm, ensuring the reasonableness of the range.
> > > > > > >
> > > > > > > > Different $\beta$ intervals for different environments
> > > > > > >
> > > > > > > Regarding Figures 11 & 12 in the appendix and your question on why different $\beta$ intervals are used for various environments: as stated in lines L630-631, FamO2O is shown to be insensitive to the range of coefficient balances **within a reasonable scope**. This scope can be assured by the second principle mentioned above. Nevertheless, **due to differences in environment characteristics** such as reward scale, sparsity/density of reward, and dynamics, **the reasonable scope for different sets of environments (like locomotion and antmaze) may vary**. **This phenomenon aligns with most offline / offline-to-online algorithms**, which often require distinct $\beta$ values (or other hyperparameters) for different environmental sets.
> > > > > > >
> > > > > > > > Enforcing valid $\beta$ values
> > > > > > >
> > > > > > > To enforce valid $\beta$ values, we employ **a $\tanh$ activation layer** as the last layer of the balance model, **rescaling the $\tanh$ output range of $(-1, 1)$ to the desired $\beta$ range**. Theoretically, clipping could also serve this purpose and may be an avenue worth exploring in future studies.
> > > > > > >
> > > > > > > ----
> > > > > > >
> > > > > > > We sincerely hope that these explanations address your queries comprehensively. Thank you once again for your thoughtful questions

---

> > > > > > > > ### Comment · Reviewer_APCr · 2023-08-18
> > > > > > > >
> > > > > > > > Thank you again for carefully addressing all my concerns - now that I better understand your work I will increase my score to a 7.

---

> > > > > > > > > ### Author Response · Authors · 2023-08-19
> > > > > > > > > **Thank you**
> > > > > > > > >
> > > > > > > > > Thank you for your thoughtful examination of our work and for the constructive feedback. We are pleased that our clarifications have addressed your concerns, and we appreciate the increased score. Your insights have been instrumental in refining our paper.

---

### Official Review · Reviewer_pp8g · 2023-07-05

**Soundness:** 3 good
**Presentation:** 3 good
**Contribution:** 2 fair
**Rating:** 6
**Confidence:** 3

**Summary:**

The paper introduces a new method to mitigate the distribution shift problem in the offline to online RL problem. The paper states the intuition that the policy should behave differently on states with different values, that is, the policy should be more conservative on high return states and exploratory on the low return states. With this intuition, the paper proposed to train a family of policies in the offline to online setting, specifically, train another "policy" to parameterize the rollout policy via the state-adaptive balance coefficient. The experiments show that FamO2O outperforms previous O2O baselines, and show via a toy experiment that FamO2O indeed leans the state-wise adaptivity, and various ablations show the importance of each design choices.

**Strengths:**

1. The experiment result is solid as it evaluates on extensive D4RL dataset with different data quality and both locomotion and maze tasks.

2. The discussion sections validate the algorithm design choices, and section 6.1 verify that the algorithm indeed learns a state-wise adaptive policy, which support the intuition and motivation of the algorithm.

**Weaknesses:**

1. The exploitation vs. exploration intuition is not brand new in the offline to online setting, there is also some work with similar intuition [1]. I believe proper comparison is required given the similarity of the intuition, although I believe training a family of policies (or conditionally parametrized the policy) seems like a slight generalization.

2. It is confusing that in the universal model training (eq. (9)), the exponential of the advantages are weighted by the balance parameter $\beta$, but when training the balance model, the Q-function (as in the loss) is not weighted by the balance parameter. There seems to be some consistency. I can tell that the unweighted objective (the current form of eq (10)) would be more computationally friendly, but theoretically, it seems more natural to optimize over a weighted version where eq. 10 is also weighted by $\beta, which sample from the balance model.

3. The action distance in eq. (11) may not be the best metric to measure the discrepancy between a policy and a trajectory. For example, if the policy that induces the trajectory only takes $a_1$ in $s$, and the evaluated policy takes $a_1$ with $p=0.51$ and $a_2$ with $p=0.49$, which may induce a very negative reward, or cause great trajectory derailment (which is not recorded in the offline trajectory), the proposed metric will still be 0 but in reality the evaluated policy is not that close to the trajectory.

**Questions:**

See above.

---

> ### Author Rebuttal · Authors · 2023-08-06
>
> Thank you for your careful review and constructive suggestions, which have helped us improve our manuscript.
>
> > Q1: The exploitation vs. exploration intuition is not brand new in the offline to online setting, there is also some work with similar intuition [1]. I believe proper comparison is required given the similarity of the intuition, although I believe training a family of policies (or conditionally parametrized the policy) seems like a slight generalization.
>
> Thank you for bringing up the important issue of exploitation vs. exploration intuition and the comparison to previous work. **Regrettably, the reference [1] was not provided in your comments, so we are unable to make a direct comparison to the specific work you are referring to.** If you could kindly provide the missing reference, we will be more than happy to provide a detailed response and comparison.
>
> In addition, we would like to highlight that Reviewer APCr has pointed out four references that bear similarity to the ideas in our paper. We have conducted a thorough comparison with those references and addressed the similarities and differences in [our response to Reviewer APCr's Q3](https://openreview.net/forum?id=vtoY8qJjTR&noteId=EQ1sNpef2S).
>
> ---
> > Q2: It is confusing that in the universal model training (eq. (9)), the exponential of the advantages are weighted by the balance parameter $\beta$, but when training the balance model, the Q-function (as in the loss) is not weighted by the balance parameter. There seems to be some consistency. I can tell that the unweighted objective (the current form of eq (10)) would be more computationally friendly, but theoretically, it seems more natural to optimize over a weighted version where eq. 10 is also weighted by $\beta$, which sample from the balance model.
>
> Thank you for your observations on Equations (9) and (10). We acknowledge your suggestion to weight Q with the balance coefficient in Equation (10).
>
> We will explore the effects of this weighting in Equation (10) in two cases:
>
> - If Q were weighted by $\beta_{\mathbf{s}} \sim \pi_b$ **without stopping the gradient** (see the equation below), training $\pi_b$ would aim to find the $\beta_{\mathbf{s}}$ that maximizes the Q-value and also increase $\beta_{\mathbf{s}}$, the output of $\pi_b$. The latter is undesired, as FamO2O's objective is to find the proper $\beta_{\mathbf{s}}$ for each state $\mathbf{s}$, not to pursue larger $\beta_{\mathbf{s}}$ leading to an aggressive policy.
>
> $\qquad\pi_b^{k+1}=\underset{\pi_b}{\arg \max}$
> $\mathbb{E}_{(s, a)\sim\mathcal{D}}[$
>
> $\qquad\quad{\color{red}\beta_{s}} Q^k(\mathbf{s}, \pi_u^{k+1}(\mathbf{s}, \beta_{\mathbf{s}}))], \quad\text{where}\quad \beta_{\mathbf{s}}\sim\pi_b(\mathbf{s}).$
>
> - If **the gradient were stopped** (see the equation below), the more aggressive balance could result in larger $\beta_{\mathbf{s}}$, skewing the Q-value. This would focus $\pi_b$ on radical policies, leading to possible extrapolation error.
>
> $\qquad\pi_b^{k+1}=\underset{\pi_b}{\arg \max}$
> $\mathbb{E}_{(s, a)\sim\mathcal{D}}[$
>
> $\qquad\quad{\color{red}\operatorname{stopgrad}(\beta_{s})} Q^k(\mathbf{s}, \pi_u^{k+1}(\mathbf{s}, \beta_{\mathbf{s}}))], \quad\text{where}\quad \beta_{\mathbf{s}}\sim\pi_b(\mathbf{s}).$
>
> In summary, the unweighted formulation in Equation (10) is intentionally chosen to let $\pi_b$ find the proper balance between improvement and constraint for maximum returns. We hope this clarifies our design decision.
>
> ---
> > Q3: The action distance in eq. (11) may not be the best metric to measure the discrepancy between a policy and a trajectory. For example, if the policy that induces the trajectory only takes $a_1$ in $s$, and the evaluated policy takes $a_1$ with $p=0.51$ and $a_2$ with $p=0.49$, which may induce a very negative reward, or cause great trajectory derailment (which is not recorded in the offline trajectory), the proposed metric will still be 0 but in reality the evaluated policy is not that close to the trajectory.
>
> Thank you for your insightful observation regarding the action distance in Equation (11). We understand the scenario you described. However, our specific formulation for action distance is defined as
>
> $d_{\text{action}}^{\pi,\tau} = \mathbb{E}_{(s, a)\sim\tau}[|| \underset{a'}{\arg\max}\pi(a'|s)-a||_2^2]$.
>
> Since the environments in D4RL are continuous action spaces, **$\pi$ outputs a normal distribution**, and thus
>
> $\arg\max_{a'}\pi(a'|s)=\mathbb{E}_{a'}[\pi(a'|s)]$.
>
> With this premise, **the action distance can provide an accurate measure of the distance between the center of the distribution and the actions within the dataset**, and it should adequately reflect the discrepancy between a policy and a trajectory in the context of our work.
>
> ---
>
> Once again, we express our gratitude for your expertise and careful consideration. We remain open to further dialogue and are eager to address any more questions or concerns you may have.

---

> > ### Comment · Reviewer_pp8g · 2023-08-17
> > **Response**
> >
> > I appreciate the authors' detailed response and my concerns (2 & 3) are addressed, and I increased my score accordingly.
> > I also apologize for not specifying the reference, and if I recall correctly, [1] should be
> >
> > [1] Zhang, Haichao, We Xu, and Haonan Yu. "Policy Expansion for Bridging Offline-to-Online Reinforcement Learning." arXiv preprint arXiv:2302.00935 (2023).

---

> > > ### Author Response · Authors · 2023-08-18
> > > **Thank you**
> > >
> > > Thank you for your considerate review and for identifying an important reference. Concerning [1], which details the method of alternating between offline policy $\pi_\beta$ and online policy $\pi_\theta$ (starting from scratch), it indeed stands as a significant and relevant work to our research. We appreciate your recommendation, and we commit to including a comparison with this work in the revised version of our paper. Your insight and guidance are gratefully acknowledged, and we thank you once again for your constructive feedback.

---

### Author Rebuttal · Authors · 2023-08-07

### References for All Reviewers:

**Dear reviewers, due to the rebuttals' character limit, we've placed the references for all rebuttals below. Thank you for your time and consideration.**

[1] Luo, Yicheng, et al. "Finetuning from Offline Reinforcement Learning: Challenges, Trade-offs and Practical Solutions." *arXiv preprint arXiv:2303.17396* (2023).

[2] Piche, Alexandre, et al. "Implicit Offline Reinforcement Learning via Supervised Learning." *arXiv preprint arXiv:2210.12272* (2022).

[3] Nikishin, Evgenii, et al. "The primacy bias in deep reinforcement learning." *International conference on machine learning*. PMLR, 2022.

[4] Lee, Seunghyun. "Offline-to-online reinforcement learning via balanced experience replay and pessimistic Q-ensemble." (2021).

[5] Hong, Joey, Aviral Kumar, and Sergey Levine. "Confidence-conditioned value functions for offline reinforcement learning." *arXiv preprint arXiv:2212.04607* (2022).

[6] Swazinna, Phillip, Steffen Udluft, and Thomas Runkler. "User-Interactive Offline Reinforcement Learning." *arXiv preprint arXiv:2205.10629* (2022).

[7] Ghosh, Dibya, et al. "Offline rl policies should be trained to be adaptive." *International Conference on Machine Learning*. PMLR, 2022.

[8] Emmons, Scott, et al. "Rvs: What is essential for offline rl via supervised learning?." *arXiv preprint arXiv:2112.10751* (2021).

[9] Chen, Lili, et al. "Decision transformer: Reinforcement learning via sequence modeling." *Advances in neural information processing systems* 34 (2021): 15084-15097.

[10] Nair, Ashvin, et al. "Awac: Accelerating online reinforcement learning with offline datasets." arXiv preprint arXiv:2006.09359 (2020).

---

### Follow-up Rebuttal for Reviewer urL3:

> Q7: Explain why you consider the term $\log\pi(a|s)$ in Eqn. (1) as a policy constraint. What is the target of this constraint, especially when $(s, a)$ is an online interaction sample?

**Explanation on policy constraint.** If we remove the policy improvement term in Equation 1, the Equation 1 will become $L_\pi = \mathbb{E}_{(\mathbf{s}, \mathbf{a})\sim\mathcal{D}}[\exp(\beta)\cdot \log\pi(\mathbf{a}|\mathbf{s})]$. As $\exp(\beta)$ is a constant that can be absorbed in the learning rate, the equation can be further simplified into

$L_\pi = \mathbb{E}_{(\mathbf{s}, \mathbf{a})\sim\mathcal{D}}[\log\pi(\mathbf{a}|\mathbf{s})]$,

which is behavior cloning that maximizes the log-likelihood of the action $\mathbf{a}$ under the state $\mathbf{s}$. Therefore, we term $\log\pi(\mathbf{a}|\mathbf{s})$ as a policy constraint due to it forcing the behavior of policy $\pi$ to be close to that of the collected dataset $\mathcal{D}$.

**Target of policy constraint.** The constraint stops excessive updates and exploration of unknown areas, particularly during the switch from offline pre-training to online fine-tuning, where it stabilizes training and prevents a performance decline (analyzed by AWAC [10]).

**Effect of policy constraint for online samples.** As new online interaction samples are added to dataset $\mathcal{D}$, they may be part of the sampled data, ensuring that the behavior policy, to which policy $\pi$ is closely constrained, gradually aligns with the online state-action distribution (also analyzed by AWAC [10]).

> Q8: Explain more on how to get Eqn. (12) and (13) from Eqn. (5)? Expand Appendix C.1 to include more details.

Thanks for your advice, and we will include more details in Appendix C.1 in the next version. Due to the character limits, please refer to the attached PDF of this global rebuttal for an explanation of how to get Eq. (12), (13) from Eq. (5).

> Q9: Miss a "$∀\epsilon\ge0$" before $\exists\{\cdots\}$ in Eqn. (6)?

Thank you for pointing this out. We will add "$\forall \epsilon \ge 0$" to Eqn. (6) in the revised version.

> Q10: [L161] Explain on the cooperation between $\pi_u$ and $\pi_b$. Why not still randomly sample $\beta_{s}$ during online fine-tuning?

Thank you for the feedback. The intuitions behind the interplay between the universal model $\pi_u$ and the balance model $\pi_b$ are as follows:
1. **Offline Pre-training Phase**: At this stage, due to the absence of online feedback, it's uncertain which balance coefficient $\beta_{\mathbf{s}}$ results in an optimal policy for any given state $\mathbf{s}$. Consequently, **$\pi_u$ is exposed to random $\beta_{\mathbf{s}}$ values, enabling it to learn from a diverse range of policies**. On the other hand, the balance model $\pi_b$ is trained by maximizing the Q-value, but the Q-value might not always be accurate, particularly for unseen areas. As such, during offline pre-training, **it's not feasible for $\pi_b$ to pinpoint the ideal balance coefficients for $\pi_u$.**
2. **Online Fine-tuning Phase**: With the advantage of online feedback, the Q-value refines, enhancing the reliability of $\pi_b$. Allowing $\pi_b$ to ascertain $\beta_{\mathbf{s}}$ for $\pi_u$ not only **benefits from $\pi_b$'s improved performance** but also **compels $\pi_u$ to emphasize the candidate policies frequently opted by $\pi_b$**.

> Q11: [L159] How many $\beta_{\mathbf{s}}$ vectors are required to learn $\pi_u$? How does this number scale with the number of states in the dataset? And how to select/design the balance coefficient space $\mathcal{B}$?

Thank you for your question concerning the number of $\beta_{\mathbf{s}}$ vectors and the selection of the balance coefficient space $\mathcal{B}$. In our paper, $\beta_{\mathbf{s}}$ is actually a scalar, sampled from $\mathcal{B} = [\beta_{\text{low}},\beta_{\text{high}}]$, thus **the number of $\beta_{\mathbf{s}}$ is infinite**.

As for the selection of $\mathcal{B}$, we have discussed this in lines L171-172, Appendix E.2 and F.2. Briefly, our method's performance is insensitive to the choice of $\beta_{\text{low}}$ and $\beta_{\text{high}}$, provided we refrain from using extremely radical values in $\mathcal{B}$. More details can be found in the aforementioned parts of the paper.

---

### Decision · Program_Chairs · 2023-09-21

**Decision:**

Accept (spotlight)

**Comment:**

This paper is concerned with addressing the distributional shift problem in offline-to-online reinforcement learning (RL) through introducing FamO2O, a framework that enables existing algorithms to determine state-adaptive improvement-constraint balances. The proposed FamO2O framework leverages a universal model to train policies with varying improvement and constraint intensities, and a balance model to select the most suitable policy for each state, emphasizing the necessity of state-adaptive balances for achieving higher performance bounds. Experiments demonstrate that FamO2O outperforms previous baselines. Generally, this paper is strong in its simple yet effective methods, with both theoretical and experimental support. The presentation is also satisfactory.